# Patient education materials for non-specific low back pain and sciatica: A systematic review and meta-analysis

**Bradley Furlong**[1]*, **Holly Etchegary**[2], **Kris Aubrey-Bassler**[1], **Michelle Swab**[3], **Andrea Pike**[1], **Amanda Hall**[1]

**1** Primary Healthcare Research Unit, Memorial University of Newfoundland, St. John's, Newfoundland and Labrador, Canada, **2** Clinical Epidemiology, Memorial University of Newfoundland, St. John's, Newfoundland and Labrador, Canada, **3** Health Sciences Library, Memorial University of Newfoundland, St. John's, Newfoundland and Labrador, Canada

* bradley.furlong@mun.ca

**Data Availability Statement:** All data are provided in the article and the Supporting information files.

**Funding:** This study is supported by the Office of Research and Graduate Studies (Medicine) at

## Abstract

### Introduction

Guidelines recommend patient education materials (PEMs) for low back pain (LBP), but no systematic review has assessed PEMs on their own. We investigated the effectiveness of PEMs on process, clinical, and health system outcomes for LBP and sciatica.

### Methods

Systematic searches were performed in MEDLINE, EMBASE, CINAHL, PsycINFO, SPORTDiscus, trial registries and grey literature through OpenGrey. We included randomized controlled trials of PEMs for LBP. Data extraction, risk of bias, and quality of evidence gradings were performed independently by two reviewers. Standardized mean differences or risk ratios and 95% confidence intervals were calculated, and effect sizes pooled using random-effects models. Analyses of acute/subacute LBP were performed separately from chronic LBP at immediate, short, medium, and long-term (6, 12, 24, and 52 weeks, respectively).

### Results

27 studies were identified. Compared to usual care for chronic LBP, we found moderate to low-quality evidence that PEMs improved pain intensity at immediate (SMD = -0.16 [95% CI: -0.29, -0.03]), short (SMD = -0.44 [95% CI: -0.88, 0.00]), medium (SMD = -0.53 [95% CI: -1.01, -0.05]), and long-term (SMD = -0.21 [95% CI: -0.41, -0.01]), medium-term disability (SMD = -0.32 [95% CI: -0.61, -0.03]), quality of life at short (SMD = -0.17 [95% CI: -0.30, -0.04]) and medium-term (SMD = -0.23 [95% CI: -0.41, -0.04]) and very low-quality evidence that PEMs improved global improvement ratings at immediate (SMD = -0.40 [95% CI: -0.58, -0.21]), short (SMD = -0.42 [95% CI: -0.60, -0.24]), medium (SMD = -0.46 [95% CI: -0.65, -0.28]), and long-term (SMD = -0.43 [95% CI: -0.61, -0.24]). We found very low-quality evidence that PEMs improved pain self-efficacy at immediate (SMD = -0.21 [95% CI: -0.39,

Memorial University of Newfoundland and the
Canadian Institute of Health Research, grant
number 398 527. The funding bodies played no
role in the design of the study; collection, analysis,
and interpretation of data; or in writing the
manuscript.

**Competing interests:** The authors have declared
that no competing interests exist.

-0.03]), short (SMD = -0.25 [95% CI: -0.43, -0.06]), medium (SMD = -0.23 [95% CI: -0.41, -0.05]), and long-term (SMD = -0.32 [95% CI: -0.50, -0.13]), and reduced medium-term fear-avoidance beliefs (SMD = -0.24 [95% CI: -0.43, -0.06]) and long-term stress (SMD = -0.21 [95% CI: -0.39, -0.03]). Compared to usual care for acute LBP, we found high to moderate-quality evidence that PEMs improved short-term pain intensity (SMD = -0.24 [95% CI: -0.42, -0.06]) and immediate-term quality of life (SMD = -0.24 [95% CI: -0.42, -0.07]). We found low to very low-quality evidence that PEMs increased knowledge at immediate (SMD = -0.51 [95% CI: -0.72, -0.31]), short (SMD = -0.48 [95% CI: -0.90, -0.05]), and long-term (RR = 1.28 [95% CI: 1.10, 1.49]) and pain self-efficacy at short (SMD = -0.78 [95% CI: -0.98, -0.58]) and long-term (SMD = -0.32 [95% CI: -0.52, -0.12]). We found moderate to very low-quality evidence that PEMs reduced short-term days off work (SMD = -0.35 [95% CI: -0.63, -0.08]), long-term imaging referrals (RR = 0.60 [95% CI: 0.41, 0.89]), and long-term physician visits (SMD = -0.16 [95% CI: -0.26, -0.05]). Compared to other interventions (e.g., yoga, Pilates), PEMs had no effect or were less effective for acute/subacute and chronic LBP.

## Conclusions

There was a high degree of variability across outcomes and time points, but providing PEMs appears favorable to usual care as we observed many small, positive patient and system impacts for acute/subacute and chronic LBP. PEMs were generally less effective than other interventions; however, no cost effectiveness analyses were performed to weigh the relative benefits of these interventions to the likely less costly PEMs.

## Introduction

Low back pain (LBP) accounts for more disability than any other musculoskeletal condition [1] and is among the five most common reasons why patients visit their family physicians [2]. It represents a substantial economic burden resulting from both direct (e.g., health care costs) and indirect costs [3] (e.g., productivity loss and compensation claims) [4, 5].

International, evidence-based guidelines for the treatment and management of LBP [6–10] recommend that for non-specific LBP (LBP that is not attributable to a recognizable, specific pathology) [11] investigations such as imaging are not required. Instead, they recommend that management should include reassurance, simple analgesics, self-care strategies, and advice and education. Patient education materials (PEMs) for LBP are intended to transfer accurate knowledge about diagnosis, prognosis, and ways to manage pain and aid recovery in order to correct false/unhelpful beliefs, reassure patients about prognosis, and manage their expectations of recovery. We hypothesized that by modifying beliefs and expectations, PEMs may reduce fear or concern related to pain, modify patients' experience of pain and expectation for unnecessary tests or other referrals, and increase patients' self-efficacy to engage in recommended strategies to manage pain which should facilitate recovery.

Indeed, Lim et al. [12] recently showed that people living with LBP want education–specifically, clear and consistent information about their LBP presented in language they can follow that includes self-management strategies and treatment options. Other systematic reviews have assessed patient education for LBP [13–25] as discussed in our protocol [26]. The most relevant review was published in 2008 [27], but variation in the education interventions of the

24 studies precluded meta-analysis limiting our understanding of the effectiveness of PEMs. Subsequent reviews have focused on clinical outcomes or broader interventions and therefore, none have fully assessed outcomes that would test our hypothesis.

This is the first systematic review and meta-analysis to investigate the effect of PEMs alone on a comprehensive set of outcomes for non-specific LBP and sciatica. The primary aim of this review is to provide up-to-date evidence on the effectiveness of these materials on immediate process outcomes such as knowledge, attitudes, and fear-avoidance beliefs; clinical outcomes such as pain and physical disability; and health system outcomes such as healthcare utilization and cost effectiveness in patients with acute and chronic non-specific LBP or sciatica.

## Methods

We published our protocol for this systematic review and meta-analysis [26].

### Search strategy

A professional librarian adapted the search strategy (S1 File) used by Engers et al. [27] which was later peer-reviewed following the Peer Review of Electronic Search Strategies (PRESS) guidelines [28]. They searched MEDLINE, EMBASE, CINAHL, PsycINFO, and SPORTDiscus from inception to March 24, 2022, as well as trial registries and grey literature using OpenGrey.

### Study selection

Results from the electronic database search were de-duplicated in Endnote [29] and imported to Covidence systematic review software [30]. Google translate was used for all non-English articles and study authors were contacted for clarification if needed. Title and abstract and full-text review were conducted by two reviewers (BF, one of GD, AS, SG; see acknowledgements) using a screening form that included pre-specified inclusion and exclusion criteria (S2 File); conflicts were resolved by a third reviewer (AH). Reference lists of relevant studies were hand-searched, and authors of conference abstracts or ongoing trials were contacted to identify additional studies. If a paper related to a study identified in a conference abstract could not be found, it was excluded.

### Data extraction

Two reviewers (BF, one of AS, SG; see acknowledgements) independently extracted data for all studies using standardized data extraction forms in Microsoft Excel, and conflicts were resolved by a third reviewer (AH). Data items included study information (authors, year of publication, country of data collection, LBP type and duration, sample size, outcome measures, study design, intervention group description, comparison group description), intervention details using the 12 variables in the TIDieR checklist [31] and outcome information (measurement tools, measurement scales, scoring methods and interpretation, means, and standard deviations).

### Risk of bias assessment

Risk of bias was assessed using the PEDro scale [32]. A study was at high risk of bias if 0–3 criteria on the scale were satisfied, moderate if 4–6 criteria were satisfied, and low if 7–10 criteria were satisfied. However, if randomization was not appropriate (e.g., quasi-randomization) or there was less than 85% follow-up, the study was considered to be at high risk of bias. PEDro scores were extracted from the PEDro database if available; otherwise, two reviewers (BF, AH)

independently assessed risk of bias for each study. Conflicts were discussed and, if necessary, reviewed with a third author (AP) to reach consensus.

## Data synthesis

We included the following contrasts:

1. PEMs alone vs. no intervention

2. PEMs alone vs. another intervention

3. PEMs + another intervention vs. the same intervention without PEMs

Analyses were conducted separately for acute/sub-acute (pain<12 weeks) and chronic (pain≥ 12 weeks) populations for all outcomes at immediate, short, medium, and long-term (defined as the closest follow-up time point to 6, 12, 24 and 52 weeks, respectively). For immediate-term follow-up only, if a study measured more than once during our defined timeframe (e.g., at both 2 weeks and 6 weeks), we chose the closest follow-up measure after the intervention was provided to get a more accurate depiction of the intervention's "immediate" effect. For other time points, if a study measured more than once within our specified timeframe, we chose the time point closest to 12, 24, or 52 weeks.

**Effectiveness analysis.**    Point estimates of effect size and 95% confidence intervals were used to estimate the treatment effect. Review Manager (RevMan) 5.4.1 (The Cochrane Collaboration) was used for the analysis [33]. Since different measurement tools were used for each outcome, we used the standardized mean difference for all analyses of continuous outcomes. Risk ratios were used for dichotomous outcomes. Where outcome data from multiple studies was pooled but the measurement scales pointed in different directions (e.g., one scale increased with disease severity while the others did not), we multiplied the point estimates by –1 to reverse the direction as described in the Cochrane handbook [34]. Where data for the same outcome were reported continuously and dichotomously between studies, we transformed dichotomous data into the SMD where possible using the methods described in the Cochrane handbook [35]. to allow for pooling of treatment effects. Otherwise, SMD and RR were reported separately. A random-effects model was used for each contrast since variation between each intervention was likely. We pooled the results if the participants, interventions, and outcomes were sufficiently homogenous, allowing for a small degree of clinical heterogeneity in the types of PEMs (e.g., content or delivery of the intervention) and populations assessed (e.g., duration of low back pain). If $I^2 > 75\%$, which represents potential for considerable statistical heterogeneity [36], we investigated both the level of clinical heterogeneity as well as the magnitude and direction of the differences in effect sizes across studies to determine if it remained reasonable to pool the results.

**Certainty of the evidence.**    To assess the level of certainty of the evidence, a summary of findings table was developed for each outcome using the Grades of Recommendation, Assessment, Development and Evaluation (GRADE) approach [37]. GRADE was assessed independently by two reviewers (BF, AH); our process for downgrading each of the five domains can be found in our published protocol [26] and in S2 File. Conflicts were discussed and, if necessary, reviewed with a third author (AP) to reach consensus.

**Sensitivity and subgroup analyses.**    Our primary analyses included all studies, but we excluded studies judged to be at high risk of bias due to concerns about the randomization process in a sensitivity analysis to determine if these studies influenced the results.

**Missing data.**    In cases where only the between group mean difference was provided in a study and we could not obtain the individual group summary data from the study's authors,

we used the generic inverse variance method to pool this data with that of the other studies [38]. A more complete explanation of missing data treatment is described in our protocol [26].

### Protocol deviations

We made minor deviations (further described in S2 File) to our published protocol [26]. Of note, due to small number of studies with physician-provided PEMs, we expanded our criteria to include studies where a member of the study's research team was responsible for providing the PEMs.

## Results

### Description of included trials (Table 1)

Of the 6435 unique records identified, 537 full texts were reviewed, and 27 included in the review (Fig 1). Most trials were conducted in the United States [39–48], followed by three in the United Kingdom [49–51], two each in Spain [52, 53], Sweden [54, 55], and Thailand [56, 57], and one each in Australia [58], Croatia [59], Finland [60], Germany [61], Iran [62], the Netherlands [63], and New Zealand [64]. One trial was conducted in both Denmark and Norway [65]. There were 21 RCTs [39–49, 51–56, 58, 59, 61, 65] and six cluster RCTs [50, 57, 60, 62–64], and participants were recruited largely through primary care [41, 42, 45–53, 59–61, 63–65]. Twelve trials included participants with acute LBP [39, 41, 49–51, 54, 55, 57, 60, 61, 63, 64] and 15 with chronic LBP [40, 42–48, 52, 53, 56, 58, 59, 62, 65]. PEMs interventions were compared to usual care in 14 studies [39, 48–51, 53, 55, 58, 60–65] and other interventions in 13 studies including Pilates [52], Yoga [45–47, 59], exercise [57], stretching [40], proprioceptive neuromuscular facilitation [56], massage [42], walking [44], chiropractic manipulation [41], and cognitive behavioral therapy [43, 54].

### Description of the interventions using the TIDieR checklist (Table 2)

PEMs were provided by physicians [48–51, 59, 61, 63, 64] or researchers [39–47, 52–58, 62, 65] via a hard copy booklet, leaflet or pamphlet [39–42, 44–52, 54, 56, 57, 59–61, 63, 64] with several newer studies using digital formats [43, 53, 55, 58, 62, 65]. PEMs content was similar across studies and included anatomy, causes of LBP, posture and movement, proper lifting techniques, exercises, how to manage flare-ups, pain management, importance of staying active, self-management strategies, and treatment options. Six studies intended to and/or measured delivery of the PEMs to the patient by audio-recording GP consultations [64], asking participants if they read the materials [42, 46, 54, 63] or recording participant activity in a mobile application [65].

### Risk of bias (Table 3)

10 studies had high risk of bias [39, 40, 49, 51, 56, 58–62], eight had moderate risk of bias [43, 44, 50, 53, 55, 57, 63, 64], and nine had low risk of bias [41, 42, 45–48, 52, 54, 65]. The most common source of bias was lack of blinding. Due to the nature of the intervention, none of the 27 included studies satisfied the criteria for blinding of subjects or providers and only nine of 27 studies reported blinding of outcome assessors. Nine of 10 high risk of bias studies [39, 40, 49, 56, 58–62] were the result of insufficient follow-up. Only one of six cluster RCTs [50, 57, 60, 62–64] adequately reported adjusting for clustering [60].

**Table 1. Study characteristics.**

| Study Year, Country | Age, M (SD) | Recruitment* | Education group (n) | Comparison (n) | Knowledge | Self-efficacy | Attitudes | General beliefs | Fear-avoidance beliefs | Catastrophizing | Coping | Anxiety | Stress | Depression | Pain | Disability | Quality of life | Global improvement | Function | Days off work | Imaging | Physician visits | Referrals | Cost | Risk of Bias |
|---|---|---|---|---|---|---|---|---|---|---|---|---|---|---|---|---|---|---|---|---|---|---|---|---|---|
| **ACUTE** | | | | | | | | | | | | | | | | | | | | | | | | | |
| Bucker 2010, DE | I:45.8 (14.3) C: 43.1 (12.4) | Primary care | Booklet^a (n=128) | Unrelated booklet (n=61) | Y | | | | Y | | | | | | | | Y | | | | | | | | High |
| Cherkin 1998, US | I:40.1 (11.2) C: 39.7 (9.4) | Primary care | Booklet (n=66) | Chiropractic manipulation (n=122) | | | | | | | | | | | Y | Y | Y | | | Y | | | | | Low |
| Darlow 2019, NZ | I:46.2 (14.5) C: 45.9 (14.4) | Primary care | Booklet (n=126) | Usual care (n=100) | | Y | | | Y | Y | | Y | | | Y | Y | | | | | | | | Y | Low |
| Irvine 2015, SE | NR | Community | Website (n=199) | Usual care (n=199) | Y | Y | | | | | | | | | | Y | Y | | | | | | | | Mod |
| Jellema 2005, NL | I:43.4 (11.1) C: 42.0 (12.0) | Primary care | Booklet (n=143) | Usual care (n=171) | | | | | Y | Y | | Y | | | Y | Y | Y | Y | | Y | | | | | Mod |
| Linton 2000, SE | I:44.0 (NR) C: 44.0 (NR) | Mixed | Booklet (n=70) | CBT (n=107) | | | | | Y | | | Y | | Y | Y | Y | | | | Y | | Y | | | Low |
| Little 2001, UK | I:42.0 (14.0) C: 47.0 (17.0) | Primary care | Booklet (n=81) | Usual care (n=78) | Y | | | | | | | | | | | Y | | | | | | | | | High |
| Lorig 2002, US | I:47.0 (11.6) C: 45.0 (0.9) | Community | Booklet, video (n=190) | Usual care (n=231) | | Y | | | | | | Y | | | Y | Y | | | | | | Y | | | High |
| Roberts 2002, UK | I:39.2 (10.9) C: 39.3 (9.7) | Primary care | Booklet (n=36) | Usual care (n=28) | Y | | | | | | | | | | Y | Y | | | | | | | | | Mod |
| Roland 1989, UK | O:38.0 (NR) | Primary care | Booklet (n=483) | Usual care (n=453) | Y | | | | | | | | | | | | | | | Y | | Y | Y | | High |
| Sihawong 2021, TH | I:40.2 (10.3) C: 41.6 (12.5) | Community | Booklet (n=20) | Exercise program (n=11) | | | | | | | | | | | Y | Y | | | Y | | | | | | Mod |
| Simula 2021, FI | I:41.4 (12.8) C: 44.6 (12.6) | Primary care | Booklet (n=215) | Usual care (n=203) | | | | | | | | | | | Y | Y | Y | | | Y | Y | Y | | | High |
| **CHRONIC** | | | | | | | | | | | | | | | | | | | | | | | | | |
| Areondonwong 2017, TH | I:35.4 (10.3) C: 36.2 (9.9) | Community | Booklet (n=21) | PNF (n=21) | | | | | | | | | | | Y | Y | Y | | | | | | | | High |
| Brodsky 2019, US | I:48.0 (10.1) C: 49.9 (8.7) | Community | Booklet (n=35) | Stretching exercise (n=43) | | | | | | | | | | | Y | Y | | | | | | | | | High |
| Cherkin 2001, US | I:43.8 (11.7) C: 45.7 (11.4) | Primary care | Booklet, videos (n=90) | Massage (n=78) | | | | | | | | | | | Y | Y | Y | | | Y | | | | | Low |
| Chiauzzi 2010, US | O: 46.1 (12.0) | Community | Digital booklet (n=105) | CBT website (n=104) | | Y | | | Y | Y | Y | Y | | Y | Y | Y | | Y | | | | | | | Mod |
| Ferrell 1997, US | I:72.7 (3.8) C:72.3 (3.4) | Mixed | Booklet (n=10) | Walking program (n=9) | | | | | | | | | | | Y | | Y | | Y | | | | | | Mod |
| Hodges 2021, AU | I:48.1 (14.0) C: 47.8 (14.1) | Community | Website (n=214) | Unguided care (n=226) | | | | | | | | | | | Y | Y | Y | | | | | | | | High |
| Kazemi 2021, IR | I:37.0 (5.7) C:37.0 (7.8) | Community | Website (n=60) | Usual care (n=60) | | | | | | | | | | | Y | Y | Y | | | | | | | | High |
| Kuvacic 2018, HR | O: 34.2 (4.52) | Primary care | Booklet (n=15) | Yoga (n=15) | | | | | | | | Y | Y | Y | Y | Y | Y | | | | | | | | High |
| Sandal 2021, DK & NO | I:48.3 (15.0) C: 46.7 (14.4) | Primary care | Mobile app (n=232) | Usual care (n=229) | | Y | | | Y | | | | | | Y | Y | Y | Y | | | | | | | Low |
| Saper 2017, US | I:44.2 (10.8) C: 46.4 (10.4) | Primary care | Booklet (n=64) | Yoga (n=127) | | | | | | | | | | | Y | Y | Y | Y | | | | | | | Low |
| Sherman 2005, US | I:45 (11) C: 44 (12) | Primary care | Booklet (n=30) | Yoga (n=36) | | | | | | | | | | | Y | Y | Y | | | | | | | | Low |
| Sherman 2011, US | I:50.8 (9.1) C: 46.6 (9.8) | Primary care | Booklet (n=45) | Yoga (n=92) | | | | | | | | | | | Y | Y | | Y | | | | | | | Low |
| Valenza 2017, ES | I:38 (12) C: 40 (16) | Primary care | Booklet (n=27) | Pilates (n=27) | | | | | | | | | | | Y | Y | | | | | | | | | Low |
| Valenzuela-Pascual 2019, ES | I:47.0 (11.1) C: 45.7 (8.8) | Primary care | Website (n=26) | Usual care (n=22) | | | | | Y | | | | | | Y | Y | | | | | | | | | Mod |

*(Continued)*

**Table 1.** (Continued)

| Study Year, Country | Age, M (SD) | Recruitment* | Education group (n) | Comparison (n) | Knowledge | Self-efficacy | Attitudes | General beliefs | Fear-avoidance beliefs | Catastrophizing | Coping | Anxiety | Stress | Depression | Pain | Disability | Quality of life | Global improvement | Function | Days off work | Imaging | Physician visits | Referrals | Cost | Risk of Bias |
|---|---|---|---|---|---|---|---|---|---|---|---|---|---|---|---|---|---|---|---|---|---|---|---|---|---|
| **Weiner 2020, US** | I:71.3 (7.5) C: 67.2 (5.5) | Primary care | Aging back clinic (n = 25) | Usual care (n = 30) | | | | | | | | | | | Y | Y | Y | | | | | | | | Low |

NR = not reported, I = intervention group, C = control group, O = overall study sample, CBT = cognitive behavioral therapy, PNF = proprioceptive neuromuscular facilitation, Mod = moderate risk of bias.

*Booklet refers to any type of written educational material such as a book, leaflet, brochure, pamphlet, etc.

⁺Recruitment refers to the location participants were recruited from (community recruitment was any recruitment not performed in a primary care family practice or emergency department setting, and mixed recruitment involved both primary care and community recruitment).

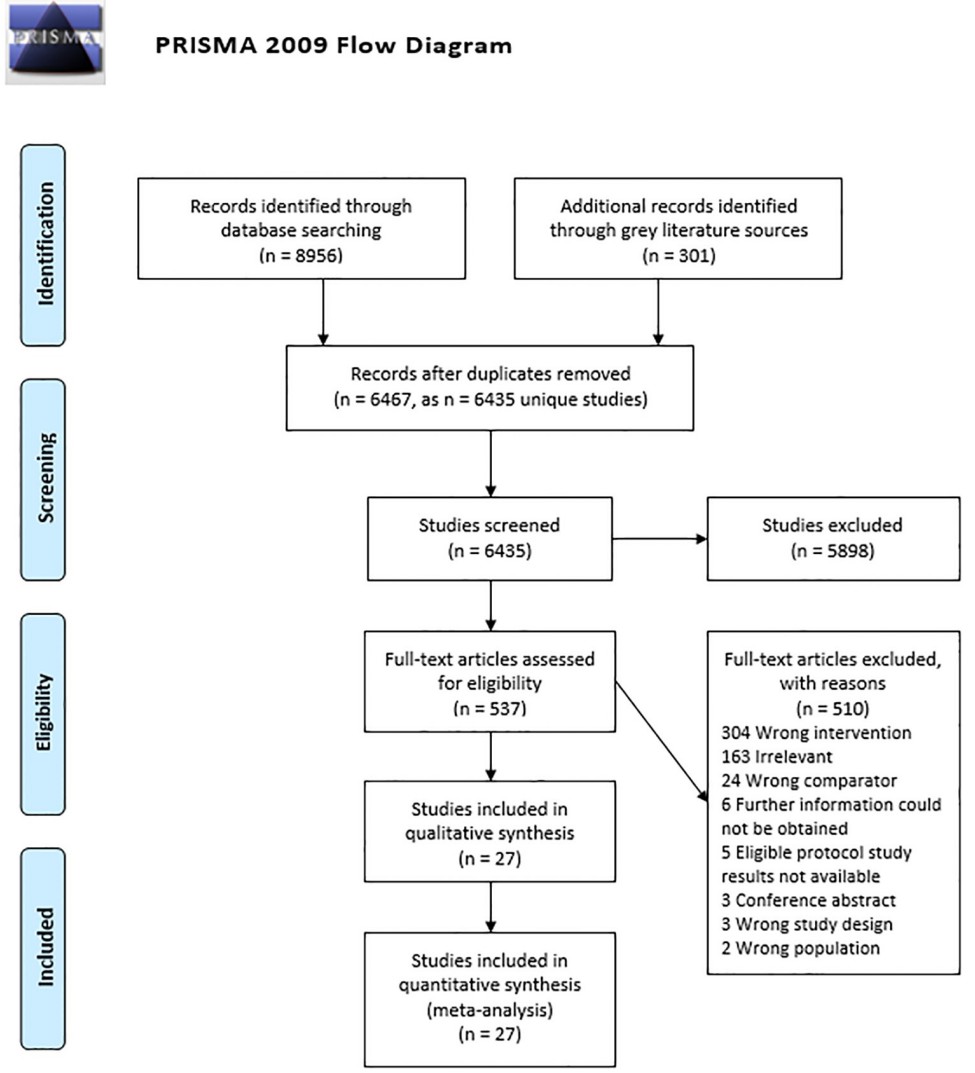

**Fig 1. PRISMA flow diagram of the systematic literature search.**

## Effectiveness of patient education materials for acute/subacute LBP

**Patient education materials alone vs. no intervention or usual care.** Nine trials [39, 49–51, 55, 60, 61, 63, 64] compared the effect of PEMs to usual care on LBP-related outcomes for acute/subacute LBP patients. In the usual care arm, patients could carry on with any LBP care as they normally would outside of the study. In one study [61], the usual care group also received a booklet with information unrelated to LBP as a control intervention. The most commonly measured outcome was disability (n = 8), followed by measures of pain intensity (n = 5), pain self-efficacy (n = 4), knowledge (n = 4), quality of life (n = 4), fear-avoidance beliefs (n = 3), catastrophizing (n = 3), anxiety (n = 3), days off work (n = 3), and physician visits (n = 3). Single studies measured global improvement, cost, imaging, and referrals. No studies measured function, general beliefs, attitudes, coping, stress, or depression. A summary of findings for eight key outcomes are presented in Table 4 (a summary of all other outcomes and forest plots for all analyses are presented in S3 and S4 Files, respectively).

Table 2. Description of the patient education material interventions using the TIDieR checklist.

| Study year | Education material | Study purpose[y] | Education content | Procedure | Mode of delivery (provider) | Consult[+] (n) | Co-interventions | Comparator description | Measured adherence/fidelity? |
|---|---|---|---|---|---|---|---|---|---|
| **ACUTE** | | | | | | | | | |
| Bucker 2010 | Booklet* | Effect of written education materials on functional capacity, fear of movement, general health, and knowledge | Booklet (NR) with information on LBP diagnosis, advice to remain active, self-management strategies | GP discussed LBP with the patient and provided the leaflet at end of consult | Face to face (GP) | Yes (1) | None | Unrelated booklet with no information about LBP | No |
| Cherkin 1998 | Booklet | Compare effect and cost of physical therapy, chiropractic manipulation, and educational booklet on LBP outcomes | Booklet (*Back in Action: A Guide to Understanding Your Low Back Pain and Learning What You Can Do About It*) with information on LBP causes, prognosis, self-management strategies, returning to normal activity, appropriate use of imaging | Booklet was mailed to participants and no further advice/consultation was provided | Mailed (researcher) | No (0) | None | Short-lever high-velocity chiropractic manipulation (up to 8 times over 4 weeks) | No |
| Darlow 2019 | Booklet | Effect and cost of consult with GP trained in FREE approach on attitudes, knowledge, confidence, and clinical behaviour | Booklet (*Free for People with Back Pain*) with information about LBP anatomy, causes, and prognosis, fear-avoidance beliefs, appropriate use of imaging, self-management strategies, returning to normal activity, acknowledgment of the difficulties of living with LBP | Booklet provided during consult with GP trained in the FREE approach (training focused on behavior change approaches to reduce provision of unhelpful LBP information) | Face to face (GP) | Yes (1) | Advice from GP trained in FREE approach | Usual care | Audio-recorded the sessions to assess FREE approach but did not report fidelity of booklet provision |
| Irvine 2015 | Website | Effect of self-management website for improving pain, quality of life, well-being, and helpful behaviours for LBP and determine correlation with behaviour change mediators | Website (*FitBack*) with both text- and video-based information on LBP, self-management and prevention strategies, and LBP exercises supported by weekly reminders and self-care messages | Participants were given access to the website at start of study (no further advice/consultation was provided) | Online (researcher) | No (0) | weekly email reminders to track pain management activities | Usual care | No |
| Jellema 2005 | Booklet | Effect of minimal intervention strategy for reducing fear-avoidance beliefs, pain catastrophizing, and distress | Booklet based on the *Back Book* (*Omgaan met lage rugpijn*) with information on LBP causes, prognosis, and treatments | Two GP consults: (1) provided advice and pain medication if necessary; (2) provided tailored information based on psychosocial prognostic factors, then provided booklet | Face to face (GP) | Yes (2) | None | Usual care | 85% of participants reported reading the booklet |
| Linton 2000 | Booklet | Effect of cognitive behavioural therapy for improving coping and reducing sick leave and healthcare utilization | Booklet (*Back Pain—Don't Suffer Needlessly*) with information on self-management strategies with an emphasis coping strategies and confronting fear-avoidance beliefs | Participants were given the booklet (no further advice/consultation was provided) | NR (researcher) | No (0) | None | Cognitive behavioral therapy (120 min sessions 1x/week for 6 weeks) | 83% of participants reported reading the booklet "word for word" at least once |
| Little 2001 | Booklet | Effect of booklet + advice on pain, function, satisfaction, and knowledge compared to pain medication + advice to stay active | Booklet (*Back Home*) with information on the LBP causes, proper lifting techniques, self-management strategies, advice to stay active and minimize bed rest, and sources for further reading | GP provided the booklet during a consult while giving supporting statements and encouragement to read the booklet | Face to face (GP) | Yes (1) | None | Usual care | No |
| Lorig 2002 | Booklet + videotape | Effect of education intervention for improving disability, pain, quality of life, role function, psychological distress, and reducing healthcare utilization | Booklet (*The Back Pain Helpbook*) and Videotape (*Easing Back: Taking Control of Your Back Problem*) with information on LBP causes, self-management strategies, flare-ups, advice to stay active, proper walking/ posture, and supportive messages from other LBP patients | Participants were given the booklet and videotape, then added to the email discussion group | NR (researcher) | No (0) | email discussion group to discuss experiences with other LBP patients and content experts | Usual care | No |
| Roberts 2002 | Booklet | Develop and test effect of booklet on knowledge, attitude, behaviour, and function | Booklet (*Back Home*) with information on the LBP causes, proper lifting techniques, self-management strategies, advice to stay active and minimize bed rest, and sources for further reading | GP provided the booklet during a consult while giving supporting statements and encouragement to read the booklet | Face to face (GP) | Yes (1) | None | Usual care | No |

(*Continued*)

**Table 2.** (Continued)

| Study year | Education material | Study purpose[y] | Education content | Procedure | Mode of delivery (provider) | Consult[+] (n) | Co-interventions | Comparator description | Measured adherence/fidelity? |
|---|---|---|---|---|---|---|---|---|---|
| Roland 1989 | Booklet | Effect of booklet on healthcare utilization and knowledge | Booklet (*Back Book*) with information on the anatomy of the back, self-management strategies, LBP exercises, how to prevent of chronification, and when to seek care | GP provided the booklet during a consult | Face to face (GP) | Yes (1) | None | Usual care | No |
| Sihawong 2021 | Booklet | Effect of risk factor education on pain and disability in office workers with neck and LBP | Booklet (NR) contained information from the *Back Book* that addressed LBP risk factors and provided information on spine function, coping with LBP, and self-management strategies | Completed a checklist of LBP risk factors, then asked to reflect on their answers using information in the booklet | Face to face (researcher) | No (0) | Completed risk factor checklist at each follow-up | Home-based stretching, strengthening, and endurance exercises (up to 5x/week) | No |
| Simula 2021 | Booklet | Effect of booklet on reducing imaging, days off work, healthcare visits, and disability, and improving function and quality of life | Booklet (*Understanding Low Back Pain*) with information on LBP causes, prevalence, self-management strategies, appropriate imaging use, advice to stay active | Provider provided booklet during a consult | Face to face (GP, physio, nurse) | Yes (1) | None | Usual care | No |
| **CHRONIC** | | | | | | | | | |
| Areeudomwong 2017 | Booklet | Effect of proprioceptive neuromuscular facilitation on pain, disability, quality of life, satisfaction, and lumbar erector spinae muscle activity | Booklet (NR) with information on LBP anatomy, causes, self-management strategies | Researcher provided the booklet, advised patients how to use it and recommended to perform exercises in the booklet | Face to face (researcher) | Yes (1) | None | Proprioceptive neuromuscular facilitation training (30 min sessions 5x/week for 4 weeks) | No |
| Brodsky 2019 | Booklet | Pilot to investigate feasibility of a larger RCT and compare data with recent similar studies | Booklet (*The Back Pain Helpbook*) with information on LBP causes, self-management strategies, managing flare-ups, importance of staying active, and targeted the role of emotions for LBP. | Researcher provided the booklet (no further advice/consultation was provided) | Face to face (researcher) | No (0) | None | Stretching exercise program (30 min sessions 1x/week for 12 weeks) + take-home stretching exercise manual | No |
| Cherkin 2001 | Booklet + videotapes | Effect and cost of acupuncture, massage, and booklet (booklet provided to control group in an effort to reduce attrition, as opposed to just providing usual care) | Booklet (*The Back Pain Helpbook*) and videotapes with information on LBP causes, self-management strategies, managing flare-ups, importance of staying active, and advice on how to cope with emotional and interpersonal problems resulting from LBP | Materials were mailed to participants (no further advice/consultation was provided) | Mailed (researcher) | No (0) | None | Soft tissue massage (60 min sessions, up to 10 sessions over 10 weeks) | 55% of participants reported reading more than 2/3 of booklet and 73% watched the videotapes |
| Chiauzzi 2010 | Booklet | Effect of cognitive behavioural therapy website for reducing distress and pain, and increasing self-efficacy, physical functioning, global impression of positive change, and use of coping strategies | Booklet (*Back Pain Guide* by the National Institute of Neurological Disorders and Stroke) with information on LBP anatomy, causes, treatment, and self-management strategies | Electronic copy of booklet emailed to participants and asked to read it over 4 weeks (no further advice/consultation was provided) | E-mailed (researcher) | No (0) | None | Cognitive-behavioural therapy website ("*painACTION*") (content provided 2x/week over 4 weeks) | No |
| Ferrell 1997 | Booklet | Effect of walking program on improving pain management for elderly people | Booklet (NR) with general information about pain and pain management | Researcher provided the booklet (no further advice/consultation was provided) | Face to face (researcher) | No (0) | Weekly telephone call (to reduce attrition) | Supervised, low-intensity walking program with stretching exercises (10–45 min sessions 4x/week over 6 weeks) | No |
| Hodges 2021 | Website | Effect of website on improving health literacy, treatment choice, and clinical outcomes compared to unguided internet use | Website (*MyBackPain*) with text- and video-based information about LBP prognosis, treatment, self-management strategies, advice to stay active, and other tailored content to increase self-efficacy and reduce negative LBP beliefs | Participants given access to website, shown how to use it, and encouraged to use it | Online (researcher) | Yes (1) | Could opt-in to emails with key messages about LBP | Self-directed LBP information seeking; asked to use the internet on their own to find information about LBP and keep diary of websites visited | No |
| Kazemi 2021 | Website | Effect of website on reducing occupational LBP in nurses compared to no intervention | Website (NR; based on the PRECEDE-PROCEED model), with information on LBP anatomy, prognosis, risk factors, exercises, ergonomics, and correct positioning of the spine | Participants given access to website and shown how to use it. Different educational topics were uploaded to the website on two separate days | Online (researcher) | Yes (1) | Weekly reminders to use website and perform exercises | Usual care | No |
| Kuvacic 2018 | Booklet | Effect of yoga and an education intervention on reducing disability, anxiety, depression, and pain | Booklet (NR) with information on LBP anatomy, ergonomics, correct posture, movement, breathing mechanisms. | GP provided booklet during consult | Face to face (GP) | Yes (1) | Newsletters (2x/ week for 8 weeks) reiterating information from booklet | Yoga (2x/week for 8 weeks) with focus on breathing techniques and emotional control | No |

*(Continued)*

**Table 2.** (Continued)

| Study year | Education material | Study purpose[¥] | Education content | Procedure | Mode of delivery (provider) | Consult[⁺] (n) | Co-interventions | Comparator description | Measured adherence/ fidelity? |
|---|---|---|---|---|---|---|---|---|---|
| **Sandal 2021** | Mobile application | Effect of mobile application on facilitating self-management of LBP, reducing disability, and improving other LBP-related outcomes | Mobile application (*selfBACK*) with general text- and video-based information about LBP, LBP exercises, self-management strategies, goal setting | Researchers provided access to the application, showed participants how to use it, and recommended using it to supplement LBP care | Online (researcher) | Yes (1) | Step counting wristband, reminders with self-management recommendations, and gamification (rewards/badges) | Usual care | 78% participants adhered to the intervention (defined as creating 6+ 'self-management plans' in the app in the first 12 weeks) |
| **Saper 2017** | Booklet | To determine if yoga is noninferior to physical therapy | Booklet (*The Back Pain Helpbook*) and videotapes with information on LBP causes, self-management strategies, managing flare-ups, importance of staying active, and advice on how to cope with emotional and interpersonal problems resulting from LBP | Researcher provided the booklet (no further advice/ consultation was provided) | NR (researcher) | No (0) | Newsletter (summarizing main points from booklet) and check-in call every 3 weeks | Yoga (75 mins, 1x/week for 12 weeks) with relaxation, meditation, and breathing techniques, and take-home yoga supplies and instructions | No |
| **Sherman 2005** | Booklet | To determine the effectiveness and safety of yoga | Booklet (*The Back Pain Helpbook*) and videotapes with information on LBP causes, self-management strategies, managing flare-ups, importance of staying active, and advice on how to cope with emotional and interpersonal problems resulting from LBP | Researcher provided the booklet (no further advice/ consultation was provided) | NR (researcher) | No (0) | None | *Viniyoga* yoga classes (75 mins, 1x/week for 12 weeks) with breathing and relaxation techniques, and take-home instructions | 100% reported reading at least part of book, 30% said they read 1/3-2/3 book, 57% reported reading more than 2/3 |
| **Sherman 2011** | Booklet | To compare the effects of yoga, stretching exercises, and self-care education | Booklet (*The Back Pain Helpbook*) and videotapes with information on LBP causes, self-management strategies, managing flare-ups, importance of staying active, and advice on how to cope with emotional and interpersonal problems resulting from LBP | Researcher provided the booklet (no further advice/ consultation was provided) | NR (researcher) | No (0) | None | *Viniyoga* yoga classes (75 mins, 1x/week for 12 weeks) with breathing and relaxation techniques, and take-home instructions | No |
| **Valenza 2017** | Booklet | Effect of Pilates on improving disability, pain, mobility, flexibility, and balance | Booklet (NR) with information on fear of movement and the importance of remaining active, postural care, lifting weights, and false beliefs. | Researcher provided the booklet (no further advice/ consultation was provided) | NR (researcher) | No (0) | None | Pilates (45 mins, 2x/week for 8 weeks) with floor exercises (using 55-cm ball) and relaxation session with rubber roller | No |
| **Valenzuela-Pascual 2019** | Website | Effect of website on decreasing pain, disability, and fear-avoidance beliefs in primary care | Website with text- and video-based information about LBP anatomy, causes, common negative LBP beliefs, appropriate imaging use, neurophysiology of pain, and pain modulation | Researcher provided access to the website (no further advice/ consultation was provided) | No (0) | Face to face (researcher) | Online discussion to share and discuss LBP experiences | Usual care | No |
| **Weiner 2020** | Booklet | Feasibility and effect of guided treatment on reducing pain and improving function | NR | Geriatrician used series of screening questionnaires to tailor treatment approach for each patient and provided booklet during consult | Face to face (geriatrician) | Yes (1) | Pre-screening questionnaire to tailor treatment | Usual care | No |

*Booklet refers to any type of written educational material such as a book, leaflet, brochure, pamphlet, or handbook.

⁺We omitted frequency and duration from the TIDieR table as the education material was provided one time in all trials. Instead, since some trials provided the education material during a consultation and others did not, we included this observation in the table along with the number of consultations held (a consultation was defined as not just the provision of the patient education material, but also verbal discussion including advice and education about LBP, how to access/used the material, or recommendations to use the material.

¥As discussed in the manuscript, the education materials were used as a control or usual care group in some studies, so the purpose of these studies may not relate to education materials.

Table 3. Risk of bias.

| Author Year | 1. Eligibility Criteria | 2. Random Allocation | 3. Allocation Concealment | 4. Similar at Baseline | 5. Blind (Subjects) | 6. Blind (Administered) | 7. Blind (Outcome Assessor) | 8. At least 85% Complete | 9. ITT Analysis | 10. Between Group Stats | 11. Point Measures/ Variability | Total Score | Risk of bias* |
|---|---|---|---|---|---|---|---|---|---|---|---|---|---|
| Areeudomwong 2017 | Y | Y | Y | Y | N | N | Y | N | N | Y | Y | 6 | High |
| Brodsky 2019 | Y | Y | N | Y | N | N | Y | N | N | Y | Y | 5 | High |
| Bucker 2010 | Y | Y | N | N | N | N | N | N | Y | Y | Y | 4 | High |
| Cherkin 1998 | Y | Y | Y | Y | N | N | Y | Y | Y | Y | Y | 8 | Low |
| Cherkin 2001 | Y | Y | N | Y | N | N | Y | Y | Y | Y | Y | 7 | Low |
| Chiauzzi 2010 | Y | Y | N | Y | N | N | N | Y | Y | Y | Y | 6 | Mod |
| Darlow 2019 | Y | Y | Y | Y | N | N | N | Y | Y | Y | N | 6 | Mod |
| Ferrell 1997 | Y | Y | N | Y | N | N | N | Y | N | Y | Y | 5 | Mod |
| Hodges 2021 | Y | Y | N | Y | N | N | Y | N | Y | Y | Y | 6 | High |
| Irvine 2015 | Y | Y | N | Y | N | N | N | Y | Y | Y | Y | 6 | Mod |
| Jellema 2005 | Y | Y | N | Y | N | N | N | Y | Y | Y | N | 5 | Mod |
| Kazemi 2021 | Y | Y | Y | Y | N | N | N | N | Y | Y | N | 4 | High |
| Kuvacic 2018 | N | Y | N | Y | N | N | N | N | N | Y | N | 4 | High |
| Linton 2000 | Y | Y | Y | Y | N | N | N | Y | Y | Y | Y | 7 | Low |
| Little 2001 | Y | Y | Y | N | N | N | N | N | N | Y | Y | 4 | High |
| Lorig 2002 | Y | Y | N | Y | N | N | N | N | Y | Y | Y | 5 | High |
| Roberts 2002 | Y | Y | Y | N | N | N | Y | Y | N | Y | N | 5 | Mod |
| Roland 1989 | Y | N | N | N | N | N | N | Y | N | Y | Y | 3 | High |
| Sandal 2021 | Y | Y | Y | Y | N | N | N | Y | Y | Y | Y | 7 | Low |
| Saper 2017 | Y | Y | N | Y | N | N | Y | Y | Y | Y | Y | 7 | Low |
| Sherman 2005 | Y | Y | Y | Y | N | N | N | Y | Y | Y | Y | 7 | Low |
| Sherman 2011 | Y | Y | Y | Y | N | N | N | Y | Y | Y | Y | 7 | Low |
| Sihawong 2021 | Y | Y | N | Y | N | N | N | Y | Y | Y | N | 5 | Mod |
| Simula 2021 | Y | Y | N | Y | N | N | N | N | N | Y | Y | 4 | High |
| Valenza 2017 | Y | Y | Y | Y | N | N | Y | Y | Y | Y | Y | 8 | Low |
| Valenzuela-Pascual 2019 | Y | Y | Y | Y | U | U | U | Y | N | Y | Y | 6 | Mod |
| Weiner 2020 | Y | Y | Y | Y | N | N | Y | Y | Y | Y | Y | 8 | Low |

* A study was deemed to have a high risk of bias if 0–3 criteria on the scale were satisfied, moderate if 4–6 criteria were satisfied, and low if 7–10 criteria were satisfied. However, if studies did not follow proper randomization methods, or did not reach 85% follow-up, we judged the study to be at high risk of bias regardless of the overall PEDro score. If cluster RCTs did not adjust for clustering we indicated this source of bias by reporting "No" for criterion #11 (Point Measures/Variability), regardless of the original judgment for this criterion.

**Table 4. Summary of findings: Education materials compared with no intervention (usual care) for acute/sub-acute low back pain.**

| Outcome (# studies) Time points | Outcome measurement tools[a] | SMD[b] (95% CI) or RR[+,-] (95% CI) | Participants (# studies) | Quality of Evidence[c] (GRADE) |
|---|---|---|---|---|
| **Knowledge (n = 5):** | | | | |
| • Immediate-term (1–8 wks) | UTs (4) | -0.51 [-0.72, -0.31] | 699 (4) | ⊕⊕⊖⊖ Low[1,4] |
| • Short-term (13–16 wks) | UTs (2) | -0.48 [-0.90, -0.05] | 502 (2) | ⊕⊕⊖⊖ Low[1,4] |
| • Medium-term | - | - | 0 (0) | No evidence |
| • Long-term (52 wks) | UTs (1) | RR[+] = 1.28 [1.10, 1.49] | 777 (1) | ⊕⊖⊖⊖ Very low[6] |
| **Self-efficacy (n = 4):** | | | | |
| • Immediate-term (2–8 wks) | PSEQ-2 (1), UTs (3) | -0.28 [-0.63, 0.07] | 650 (3) | ⊕⊕⊕⊖ Moderate[4] |
| • Short-term (16 wks) | UTs (1) | -0.78 [-0.98, -0.58] | 398 (1) | ⊕⊖⊖⊖ Very low[6] |
| • Medium-term | - | - | 0 (0) | No evidence |
| • Long-term (52 wks) | UTs (1) | -0.32 [-0.52, -0.12] | 421 (1) | ⊕⊖⊖⊖ Very low[6] |
| **Pain (n = 5):** | | | | |
| • Immediate-term (2–8 wks) | NRS (2), UTs (1) | -0.13 [-0.27, 0.01] | 910 (3) | ⊕⊕⊕⊕ High |
| • Short-term (12–16 wks) | NRS (3), UTs (1) | -0.24 [-0.42, -0.06] | 1101 (4) | ⊕⊕⊕⊕ High |
| • Medium-term (26 wks) | NRS (2) | -0.03 [-0.20, 0.15] | 515 (2) | ⊕⊕⊕⊕ High |
| • Long-term (52 wks) | NRS (2), VNS (1) | -0.11 [-0.24, 0.02] | 892 (3) | ⊕⊕⊕⊖ Moderate[1] |
| **Disability (n = 8):** | | | | |
| • Immediate-term (1–8 wks) | RMDQ (2), ALBDS (2), FFbH-R (1), WLQ (1) | -0.05 [-0.17, 0.06] | 1220 (6) | ⊕⊕⊕⊕ High |
| • Short-term (13–16 wks) | RMDQ (2), ALBDS (1), FFbH-R (1), WLQ (1), ODI (1) | -0.06 [-0.18, 0.05] | 1272 (6) | ⊕⊕⊕⊕ High |
| • Medium-term (26 wks) | RMDQ (2), ALBDS (1) | 0.09 [-0.08, 0.27] | 563 (3) | ⊕⊕⊕⊕ High |
| • Long-term (52 wks) | RMDQ (2), ALBDS (1), ODI (1) | -0.09 [-0.27, 0.08] | 938 (4) | ⊕⊕⊕⊖ Moderate[1] |
| **Quality of Life (n = 4):** | | | | |
| • Immediate-term (1–8 wks) | SF-36 (1), Dartmouth CO-OP (1) | -0.24 [-0.42, -0.07] | 524 (2) | ⊕⊕⊕⊖ Moderate[4] |
| • Short-term (13–16 wks) | SF-36 (1), Dartmouth CO-OP (1), UTs (1) | -0.20 [-0.43, 0.03] | 804 (3) | ⊕⊕⊕⊕ High |
| • Medium-term (26 wks) | UTs (1) | 0.00 [-0.23, 0.23] | 286 (1) | ⊕⊖⊖⊖ Very low[6] |
| • Long-term (52 wks) | EQ5D-3L (1), UTs (1) | 0.01 [-0.17, 0.19] | 470 (2) | ⊕⊕⊕⊖ Moderate[1] |
| **Global improvement (n = 1):** | | | | |
| • Immediate-term (6 wks) | UTs (1) | RR[-] = 1.07 [0.80, 1.43] | 305 (1) | ⊕⊖⊖⊖ Very low[6] |
| • Short-term (13 wks) | UTs (1) | RR[-] = 1.03 [0.75, 1.42] | 305 (1) | ⊕⊖⊖⊖ Very low[6] |
| • Medium-term (26 wks) | UTs (1) | RR[-] = 1.05 [0.75, 1.47] | 299 (1) | ⊕⊖⊖⊖ Very low[6] |

(*Continued*)

**Table 4.** (Continued)

| Outcome (# studies) Time points | Outcome measurement tools[a] | SMD[b] (95% CI) or RR[+,-] (95% CI) | Participants (# studies) | Quality of Evidence[c] (GRADE) |
|---|---|---|---|---|
| • Long-term (52 wks) | UTs (1) | RR[-] = 1.15 [0.81, 1.65] | 288 (1) | ⊕⊖⊖⊖ Very low[6] |
| **Days off work (n = 3):** | | | | |
| • Immediate-term (6 wks) | % with days off work (1) | RR[-] = 0.83 [0.49, 1.42] | 248 (1) | ⊕⊖⊖⊖ Very low[6] |
| • Short-term (13 wks) | % with days off work (1), mean days off work (1) | -0.35 [-0.63, -0.08] | 612 (2) | ⊕⊕⊖⊖ Low[1,4] |
| • Medium-term (26 wks) | % with days off work (1) | RR[-] = 0.33 [0.10, 1.16] | 244 (1) | ⊕⊖⊖⊖ Very low[6] |
| • Long-term (52 wks) | % with days off work (1), mean days off work (2) | -0.10 [-0.32, 0.12] | 1535 (3) | ⊕⊕⊕⊖ Moderate[1] |
| **Imaging (n = 1):** | | | | |
| • Immediate-term | - | - | 0 (0) | No evidence |
| • Short-term (13 wks) | % receiving LBP imaging (1) | RR[-] = 0.64 [0.38, 1.09] | 364 (1) | ⊕⊖⊖⊖ Very low[6] |
| • Medium-term | - | - | 0 (0) | No evidence |
| • Long-term (52 wks) | % receiving LBP imaging (1) | RR[-] = 0.60 [0.41, 0.89] | 364 (1) | ⊕⊖⊖⊖ Very low[6] |

[a]See legend in S3 File for a complete list of non-abbreviated names of all measurement tools.

[b]Data are presented as standardized mean differences (SMD) and 95% confidence intervals (95% CI) unless otherwise indicated (negative SMD favors education materials). Risk ratios are indicated with RR[+] (RR > 1 favors education) and RR[-] (RR < 1 favors education).

[c]Quality of evidence was downgraded for risk of bias,

[1] imprecision,

[2] inconsistency,

[3] indirectness,

[4] publication bias,

[5] or downgraded to very low if there was one study

[6] (more details provided in S3 File).

*Pain intensity (n = 5).* We found high-quality evidence that PEMs were significantly more effective for reducing pain intensity compared to usual care at short-term (4 RCTs, $n$ = 1101; SMD = -0.24; 95% CI: -0.42, -0.06; $p$ = 0.01; $I^2$ = 55%). We found high-quality evidence that PEMs had no effect on pain intensity compared to usual care at immediate (3 RCTs, $n$ = 910; SMD = -0.13; 95% CI: -0.27, 0.01; $p$ = 0.07; $I^2$ = 14%) and medium-term (2 RCTs, $n$ = 515; SMD = -0.03 95% CI: -0.20, 0.15; $p$ = 0.77; $I^2$ = 0%), and moderate-quality evidence of no effect at long-term (3 RCTs, $n$ = 892; SMD = -0.11; 95% CI: -0.24, 0.02; $p$ = 0.11; $I^2$ = 0%).

*Disability (n = 8).* We found high-quality evidence that PEMs had no effect on disability compared to usual care at immediate (6 RCTs, $n$ = 1220; SMD = -0.05; 95% CI: -0.17, 0.06; $p$ = 0.35; $I^2$ = 0%), short (6 RCTs, $n$ = 1272; SMD = -0.06; 95% CI: -0.18, 0.05; $p$ = 0.30; $I^2$ = 7%), and medium-term (3 RCTs, $n$ = 563; SMD = 0.09; 95% CI: -0.08, 0.27; $p$ = 0.31; $I^2$ = 6%) and moderate-quality evidence of no effect at long-term (4 RCTs, $n$ = 938; SMD = -0.09; 95% CI: -0.27, 0.08; $p$ = 0.28; $I^2$ = 37%).

*Quality of life (n = 4).* We found moderate-quality evidence that PEMs are significantly more effective than usual care for improving quality of life at immediate-term (2 RCTs, $n$ = 524; SMD = -0.24; 95% CI: -0.42, -0.07; $p$ = 0.006; $I^2$ = 0%). We found high-quality evidence that PEMs had no effect on quality of life compared to usual care at short-term (3 RCTs,

$n$ = 804; SMD = -0.20; 95% CI: -0.43, 0.03; $p$ = 0.09; $I^2$ = 58%). We found very low-quality evidence of no effect at medium-term (1 RCT, $n$ = 286; SMD = 0.00; 95% CI: -0.23, 0.23; $p$ = 1.00) and moderate-quality evidence of no effect at long-term (2 RCTs, $n$ = 470; SMD = 0.01; 95% CI: -0.17, 0.19; $p$ = 0.94; $I^2$ = 0%).

*Global improvement (n = 1)*. We found very low-quality evidence that PEMs had no effect compared to usual care on global improvement at immediate (1 RCT, $n$ = 305; RR = 1.07; 95% CI: 0.80, 1.43; $p$ = 0.64), short (1 RCT, $n$ = 305; RR = 1.03; 95% CI: 0.75, 1.42; $p$ = 0.85), medium (1 RCT, $n$ = 299; RR = 1.05; 95% CI: 0.75, 1.47; $p$ = 0.76), and long-term (1 RCT, $n$ = 288; RR = 1.15; 95% CI: 0.81, 1.65; $p$ = 0.43), where RR > 1 favors usual care.

*Knowledge (n = 5)*. We found low-quality evidence that PEMs are significantly more effective than usual care for improving knowledge in the immediate (4 RCTs, $n$ = 699; SMD = -0.51; 95% CI: -0.72, -0.31; $p$ < 0.00001; $I^2$ = 47%) and short-term (2 RCTs, $n$ = 502; SMD = -0.48; 95% CI: -0.90, -0.05; $p$ = 0.03; $I^2$ = 71%). We found very low-quality evidence that PEMs are significantly more effective than usual care for improving long-term knowledge (1 RCT, $n$ = 777; RR = 1.28; 95% CI: 1.10, 1.49; $p$ = 0.001).

*Pain self-efficacy (n = 4)*. We found moderate quality evidence that PEMs had no effect on pain self-efficacy compared to usual care at immediate-term (3 RCTs, $n$ = 650; SMD = -0.28; 95% CI: -0.63, 0.07; $p$ = 0.12; $I^2$ = 73%). We found very low-quality evidence that PEMs are significantly more effective than usual care for improving self-efficacy at short (1 RCT, $n$ = 398; SMD = -0.78; 95% CI: -0.98, -0.58; $p$ < 0.00001) and long-term (1 RCT, $n$ = 421; SMD = -0.32; 95% CI: -0.52, -0.12; $p$ = 0.002).

*Fear-avoidance beliefs (n = 3)*. We found high quality evidence that PEMs had no effect on fear-avoidance beliefs compared to usual care at immediate-term (3 RCTs, $n$ = 611; SMD = -0.14; 95% CI: -0.36, 0.09; $p$ = 0.23; $I^2$ = 44%), and very low-quality evidence of no effect at short (1 RCT, $n$ = 114; SMD = 0.00; 95% CI: -0.38, 0.38; $p$ = 1.00) and long-term (1 RCT, $n$ = 150; SMD = 0.10; 95% CI: -0.15, 0.35; $p$ = 0.43).

*Catastrophizing (n = 3)*. We found high quality evidence that PEMs had no effect on catastrophizing compared to usual care at immediate-term (3 RCTs, $n$ = 879; SMD = -0.01; 95% CI: -0.22, 0.20; $p$ = 0.92; $I^2$ = 60%), and very low-quality evidence of no effect at short (1 RCT, $n$ = 398; SMD = -0.12; 95% CI: -0.31, 0.07; $p$ = 0.22) and long-term (1 RCT, $n$ = 248; SMD = 0.07; 95% CI: -0.18, 0.32; $p$ = 0.58).

*Anxiety (n = 3)*. We found moderate-quality evidence that PEMs had no effect on anxiety compared to usual care at immediate-term (2 RCTs, $n$ = 485; SMD = -0.01; 95% CI: -0.45, 0.43; $p$ = 0.98; $I^2$ = 83%) and low-quality evidence of no effect at long-term (2 RCTs, $n$ = 673; SMD = -0.13; 95% CI: -0.52, 0.26; $p$ = 0.53; $I^2$ = 85%).

*Days off work (n = 3)*. We found low-quality evidence that PEMs were significantly more effective for reducing days off work compared to usual care at short-term (2 RCTs, $n$ = 612; SMD = -0.35; 95% CI: -0.63, -0.08; $p$ = 0.01; $I^2$ = 22%). We found very low-quality evidence that PEMs had no effect on days off work compared to usual care at immediate (1 RCT, $n$ = 248; RR = 0.83; 95% CI: 0.49, 1.42; $p$ = 0.50) and medium-term (1 RCT, $n$ = 244; RR = 0.33; 95% CI: 0.10, 1.16; $p$ = 0.08) and moderate-quality evidence of no effect at long-term (3 RCTs, $n$ = 1535; SMD = -0.10; 95% CI: -0.32, 0.12; $p$ = 0.37; $I^2$ = 62%). Sensitivity analysis for long-term follow-up revealed no difference when removing one study [51] due to concerns about their randomization method (SMD = -0.23; 95% CI: -0.46, 0.00; $p$ = 0.05; $I^2$ = 11%).

*Imaging (n = 1)*. We found very low-quality evidence that PEMs are significantly more effective for reducing imaging for LBP compared to usual care at long-term (1 RCT, $n$ = 364; RR = 0.60; 95% CI: 0.41, 0.89; $p$ = 0.01). We found very low-quality evidence that PEMs had no effect on imaging compared to usual care at short-term (1 RCT, $n$ = 364; RR = 0.64; 95% CI: 0.38, 1.09; $p$ = 0.10).

*Physician visits (n = 3)*. We found moderate-quality evidence that PEMs are significantly more effective for reducing physician visits compared to usual care at long-term (3 RCTs, *n* = 1721; SMD = -0.16; 95% CI: -0.26, -0.05; *p* = 0.003; $I^2$ = 0%). We found very low-quality evidence of no effect at short-term (1 RCT, *n* = 364; SMD = -0.07; 95% CI: -0.27, 0.13; *p* = 0.49). Sensitivity analysis for long-term follow-up revealed no difference when removing one study [51] due to concerns about their randomization method (SMD = -0.16; 95% CI: -0.31, -0.02; *p* = 0.03; $I^2$ = 0%).

*Referrals (n = 1)*. We found very low-quality evidence that PEMs are significantly more effective than usual care for reducing specialist referrals at long-term (1 RCT; *n* = 936; RR = 0.85; 95% CI: 0.58, 1.23; *p* = 0.38).

*Cost (n = 1)*. We found very low-quality evidence that PEMs had no effect on cost compared to usual care at medium-term (1 RCT, *n* = 226; SMD = -0.11; 95% CI: -0.37, 0.16; *p* = 0.43).

**Patient education materials alone vs. other interventions.** Three trials [41, 54, 57] compared the effect of PEMs to other interventions on LBP-related outcomes for acute/subacute LBP patients. The comparator interventions were cognitive behavioural therapy [54], chiropractic manipulation [41], and an exercise program [57]. The studies included measures of pain intensity (n = 3), disability (n = 3), and days off work (n = 2), and one study measured fear-avoidance beliefs, catastrophizing, anxiety, depression, and physician visits. No studies measured quality of life, global improvement, function, knowledge, self-efficacy, attitudes, general beliefs, coping, stress, imaging, referrals, or cost. A summary of findings for eight key outcomes are presented in Table 5 (a summary of all other outcomes and forest plots for all analyses are presented in S3 and S4 Files, respectively).

*Pain intensity (n = 3)*. We found very low-quality evidence that PEMs are more effective for reducing pain intensity compared to other interventions at medium-term (1 RCT, *n* = 31; SMD = -0.89; 95% CI: -1.66, -0.11; *p* = 0.02). We found very low-quality evidence that PEMs are less effective than other interventions at immediate-term (1 RCT, *n* = 178; SMD = 0.51; 95% CI: 0.20, 0.83; *p* = 0.001), low-quality evidence that PEMs have no effect on pain intensity when compared to other interventions at short-term (2 RCTs, *n* = 212; SMD = 0.07; 95% CI: -0.81, 0.95; *p* = 0.88; $I^2$ = 79%), and very low-quality evidence of no effect at long-term (1 RCT, *n* = 155; SMD = 0.04; 95% CI: -0.28, 0.36; *p* = 0.81).

*Disability (n = 3)*. We found very low-quality evidence that PEMs had no effect on disability compared to other interventions at immediate (1 RCT, *n* = 178; SMD = 0.27; 95% CI: -0.04, 0.58; *p* = 0.09) and medium-term (1 RCT, *n* = 31; SMD = -0.15; 95% CI: -0.88, 0.58; *p* = 0.69), moderate-quality evidence of no effect at short-term (2 RCTs, *n* = 212; SMD = 0.23; 95% CI: -0.06, 0.51; *p* = 0.12; $I^2$ = 0%), and low-quality evidence of no effect at long-term (2 RCTs, *n* = 343; SMD = 0.20; 95% CI: -0.04, 0.43; *p* = 0.10; $I^2$ = 0%).

*Fear-avoidance beliefs (n = 1)*. We found very low-quality evidence that PEMs had no effect on fear-avoidance beliefs compared to other interventions at long-term (1 RCT, *n* = 155; SMD = 0.17; 95% CI: -0.16, 0.49; *p* = 0.31).

*Catastrophizing (n = 1)*. We found very low-quality evidence that PEMs had no effect on catastrophizing compared to other interventions at long-term (1 RCT, *n* = 155; SMD = -0.06; 95% CI: -0.38, 0.27; *p* = 0.73).

*Anxiety (n = 1)*. We found very low-quality evidence that PEMs had no effect on anxiety compared to other interventions at long-term (1 RCT, *n* = 155; SMD = -0.05; 95% CI: -0.37, 0.27; *p* = 0.74).

*Depression (n = 1)*. We found very low-quality evidence that PEMs had no effect on depression compared to other interventions at long-term (1 RCT, *n* = 155; SMD = 0.00; 95% CI: -0.32, 0.32; *p* = 1.00).

**Table 5. Summary of findings: Education materials compared with another intervention for acute/subacute low back pain.**

| Outcome (# studies) Time points | Outcome measurement tools[a] | SMD[b] (95% CI) or RR[+,-] (95% CI) | Participants (# studies) | Quality of Evidence[c] (GRADE) |
|---|---|---|---|---|
| **Knowledge: no evidence** | | | | |
| **Self-Efficacy: no evidence** | | | | |
| **Pain (n = 3):** | | | | |
| • Immediate-term (4 wks) | SBS (1) | 0.51 [0.20, 0.83] | 178 (1) | ⊕⊖⊖⊖ Very low[6] |
| • Short-term (12 wks) | VAS (1), SBS (1) | 0.07 [-0.81, 0.95] | 212 (2) | ⊕⊕⊖⊖ Low[2,3] |
| • Medium-term (26 wks) | VAS (1) | -0.89 [-1.66, -0.11] | 31 (1) | ⊕⊖⊖⊖ Very low[6] |
| • Long-term (52 wks) | OEQ (1) | 0.04 [-0.28, 0.36] | 155 (1) | ⊕⊖⊖⊖ Very low[6] |
| **Disability (n = 3):** | | | | |
| • Immediate-term (4 wks) | RMDQ (1) | 0.27 [-0.04, 0.58] | 178 (1) | ⊕⊖⊖⊖ Very low[6] |
| • Short-term (12 wks) | RMDQ (2) | 0.23 [-0.06, 0.51] | 212 (2) | ⊕⊕⊕⊖ Moderate[2] |
| • Medium-term (26 wks) | RMDQ (1) | -0.15 [-0.88, 0.58] | 31 (1) | ⊕⊖⊖⊖ Very low[6] |
| • Long-term (48–52 wks) | ADLQ (1), % with reduced activity (1) | 0.20 [-0.04, 0.43] | 343 (2) | ⊕⊕⊖⊖ Low[2,4] |
| **Quality of Life: no evidence** | | | | |
| **Global Improvement: no evidence** | | | | |
| **Days off work (n = 2):** | | | | |
| • Immediate-term | - | - | 0 (0) | No evidence |
| • Short-term | - | - | 0 (0) | No evidence |
| • Medium-term | - | - | 0 (0) | No evidence |
| • Long-term (48–52 wks) | % with days off work (1), mean days off work (1) | 0.36 [0.09, 0.63] | 343 (2) | ⊕⊕⊖⊖ Low[2,4] |
| **Imaging: no evidence** | | | | |

[a]See legend in S3 File for a complete list of non-abbreviated names of all measurement tools.

[b]Data are presented as standardized mean differences (SMD) and 95% confidence intervals (95% CI) unless otherwise indicated (negative SMD favors education materials). Risk ratios are indicated with RR[+] (RR > 1 favors education) and RR[-] (RR < 1 favors education).

[c]Quality of evidence was downgraded for risk of bias,

[1] imprecision,

[2] inconsistency,

[3] indirectness,

[4] publication bias,

[5] or downgraded to very low if there was one study

[6] (more details provided in S3 File).

*Days off work (n = 2).* We found low-quality evidence that PEMs are significantly less effective than other interventions for reducing days off work at long-term (2 RCTs, $n = 343$; SMD = 0.36; 95% CI: 0.09, 0.63; $p = 0.01$; $I^2 = 0\%$).

*Physician visits (n = 1).* We found very low-quality evidence that PEMs were less effective than other interventions on reducing physician visits (1 RCT, $n = 155$; SMD = 0.53; 95% CI: 0.20, 0.85; $p = 0.002$) at long-term.

**Intervention vs. intervention + patient education materials (additive effect).** No studies measured the additive effect of PEMs with other interventions.

## Effectiveness of patient education materials for chronic LBP

**Patient education materials alone vs. no intervention or usual care.** Five trials [48, 53, 58, 62, 65] compared the effect of PEMs to usual care on LBP-related outcomes for chronic LBP patients. A protocol for usual care was not described in four of these studies; rather, patients could continue any LBP care as they normally would outside of the study. In one study [58], the comparator group was unguided internet use where participants were asked to seek out information about LBP on their own; we considered this similar to usual care. Outcomes measured included pain intensity (n = 5), disability (n = 5), quality of life (n = 4), fear-avoidance beliefs (n = 2), and one study measured global improvement, self-efficacy, stress, and depression. No studies measured function, knowledge, attitudes, general beliefs, catastrophizing, coping, anxiety, days off work, imaging, physician visits, referrals, or cost. A summary of findings for eight key outcomes are presented in Table 6 (a summary of all other outcomes and forest plots for all analyses are presented in S3 and S4 Files, respectively).

*Pain intensity (n = 5).* We found moderate-quality evidence that PEMs were significantly more effective for reducing pain intensity compared to usual care at immediate (4 RCTs, n = 890; SMD = -0.16; 95% CI: -0.29, -0.03; $p = 0.02$; $I^2 = 0\%$) and long-term (2 RCTs, n = 757; SMD = -0.21; 95% CI: -0.41, -0.01; $p = 0.04$; $I^2 = 47\%$), and low-quality evidence of the same observation at short (4 RCTs, n = 925; SMD = -0.44; 95% CI: -0.88, 0.00; $p = 0.05$; $I^2 = 89\%$) and medium-term (4 RCTs, n = 907; SMD = -0.53; 95% CI: -1.01, -0.05; $p = 0.03$; $I^2 = 90\%$).

*Disability (n = 5).* We found moderate-quality evidence that PEMs are significantly more effective for reducing disability compared to usual care at medium-term (4 RCTs, n = 939; SMD = -0.32; 95% CI: -0.61, -0.03; $p = 0.03$; $I^2 = 74\%$). We found moderate-quality evidence of no effect at immediate (4 RCTs, n = 919; SMD = -0.12; 95% CI: -0.31, 0.07; $p = 0.23$; $I^2 = 38\%$), short (4 RCTs, n = 964; SMD = -0.23; 95% CI: -0.48, 0.03; $p = 0.08$; $I^2 = 68\%$), and long-term (2 RCT, n = 770; SMD = -0.12; 95% CI: -0.27, 0.02; $p = 0.09$; $I^2 = 0\%$).

*Quality of life (n = 4).* We found moderate-quality evidence that PEMs are significantly more effective for increasing quality of life compared to usual care at short (4 RCTs, n = 934; SMD = -0.15; 95% CI: -0.28, -0.03; $p = 0.02$; $I^2 = 0\%$) and medium-term (4 RCT, n = 902; SMD = -0.23; 95% CI: -0.41, -0.04; $p = 0.02$; $I^2 = 39\%$). We found moderate-quality evidence of no effect at immediate (3 RCT, n = 839; SMD = -0.04; 95% CI: -0.18, 0.09; $p = 0.55$; $I^2 = 0\%$) and long-term (2 RCT, n = 748; SMD = -0.13; 95% CI: -0.28, 0.01; $p = 0.07$; $I^2 = 0\%$).

*Global improvement (n = 1).* We found very low-quality evidence that PEMs were significantly more effective at increasing global improvement ratings compared to usual care at immediate (1 RCT, n = 461; SMD = -0.40; 95% CI: -0.58, -0.21; $p < 0.0001$), short (1 RCT, n = 461; SMD = -0.42; 95% CI: -0.60, -0.24; $p < 0.00001$), medium (1 RCT, n = 461; SMD = -0.46; 95% CI: -0.65, -0.28; $p < 0.00001$), and long-term (1 RCT, n = 461; SMD = -0.43; 95% CI: -0.61, -0.24; $p < 0.00001$).

*Self-efficacy (n = 1).* We found very low-quality evidence that PEMs were significantly more effective at increasing self-efficacy compared to usual care at immediate (1 RCT, n = 461; SMD = -0.21; 95% CI: -0.39, -0.03; $p = 0.02$), short (1 RCT, n = 461; SMD = -0.25; 95% CI: -0.43, -0.06; $p = 0.009$), medium (1 RCT, n = 461; SMD = -0.23; 95% CI: -0.41, -0.05; $p = 0.01$), and long-term (1 RCT, n = 461; SMD = -0.32; 95% CI: -0.50, -0.13; $p = 0.0007$).

*Fear-avoidance beliefs (n = 2).* We found very low-quality evidence that PEMs were significantly more effective for reducing fear-avoidance beliefs compared to usual care at medium-term (1 RCT, n = 461; SMD = -0.24; 95% CI: -0.43, -0.06; $p = 0.01$). We found high-quality

**Table 6. Summary of findings: Education materials compared with no intervention (usual care) for chronic low back pain.**

| Outcome (# studies) Time points | Outcome measurement tools[a] | SMD[b] (95% CI) or RR[+,-] (95% CI) | Participants (# studies) | Quality of Evidence[c] (GRADE) |
|---|---|---|---|---|
| **Knowledge: no evidence** | | | | |
| **Self-Efficacy (n = 1):** | | | | |
| • Immediate (6 wks) | PSEQ (1) | -0.21 [-0.39, -0.03] | 461 (1) | ⊕⊖⊖⊖ Very low[6] |
| • Short-term (13 wks) | PSEQ (1) | -0.25 [-0.43, -0.06] | 461 (1) | ⊕⊖⊖⊖ Very low[6] |
| • Medium-term (26 wks) | PSEQ (1) | -0.23 [-0.41, -0.05] | 461 (1) | ⊕⊖⊖⊖ Very low[6] |
| • Long-term (39 wks) | PSEQ (1) | -0.32 [-0.50, -0.13] | 461 (1) | ⊕⊖⊖⊖ Very low[6] |
| **Pain (n = 5):** | | | | |
| • Immediate (2–6 wks) | VAS (2), NRS (1), UTs (1) | -0.16 [-0.29, -0.03] | 890 (4) | ⊕⊕⊕⊖ Moderate[1] |
| • Short-term (12–13 wks) | VAS (2), NRS (1), UTs (1) | -0.44 [-0.88, 0.00] | 925 (4) | ⊕⊕⊖⊖ Low[1,3] |
| • Medium-term (24–26 wks) | VAS (2), NRS (1), UTs (1) | -0.53 [-1.01, -0.05] | 907 (4) | ⊕⊕⊖⊖ Low[1,3] |
| • Long-term (39–52 wks) | VAS (1), NRS (1) | -0.21 [-0.41, -0.01] | 757 (2) | ⊕⊕⊕⊖ Moderate[1] |
| **Disability (n = 5):** | | | | |
| • Immediate (2–6 wks) | RMDQ (4) | -0.12 [-0.31, 0.07] | 919 (4) | ⊕⊕⊕⊖ Moderate[1] |
| • Short-term (12–13 wks) | RMDQ (3), QBPDS (1) | -0.23 [-0.48, 0.03] | 964 (4) | ⊕⊕⊕⊖ Moderate[1] |
| • Medium-term (24–26 wks) | RMDQ (3), QBPDS (1) | -0.32 [-0.61, -0.03] | 939 (4) | ⊕⊕⊕⊖ Moderate[1] |
| • Long-term (39–52 wks) | RMDQ (2) | -0.12 [-0.27, 0.02] | 770 (2) | ⊕⊕⊕⊖ Moderate[1] |
| **Quality of Life (n = 4):** | | | | |
| • Immediate (4–6 wks) | AQoL-8D (1), SF-12 (1), EQ-5D (1) | -0.04 [-0.18, 0.09] | 839 (3) | ⊕⊕⊕⊖ Moderate[1] |
| • Short-term (12–13 wks) | AQoL-8D (1), SF-12 (1), SF-36 (1), EQ-5D (1) | -0.15 [-0.28, -0.03] | 934 (4) | ⊕⊕⊕⊖ Moderate[1] |
| • Medium-term (24–26 wks) | AQoL-8D (1), SF-12 (1), SF-36 (1), EQ-5D (1) | -0.23 [-0.41, -0.04] | 902 (4) | ⊕⊕⊕⊖ Moderate[1] |
| • Long-term (39–52 wks) | AQoL-8D (1), EQ-5D (1) | -0.13 [-0.28, 0.01] | 748 (2) | ⊕⊕⊕⊖ Moderate[1] |
| **Global Improvement** | | | | |
| • Immediate (6 wks) | GPE (1) | -0.40 [-0.58, -0.21] | 461 (1) | ⊕⊖⊖⊖ Very low[6] |
| • Short-term (13 wks) | GPE (1) | -0.42 [-0.60, -0.24] | 461 (1) | ⊕⊖⊖⊖ Very low[6] |
| • Medium-term (26 wks) | GPE (1) | -0.46 [-0.65, -0.28] | 461 (1) | ⊕⊖⊖⊖ Very low[6] |
| • Long-term (39 wks) | GPE (1) | -0.43 [-0.61, -0.24] | 461 (1) | ⊕⊖⊖⊖ Very low[6] |
| **Days off work: no evidence** | | | | |

(*Continued*)

**Table 6.** (Continued)

| Outcome (# studies) Time points | Outcome measurement tools[a] | SMD[b] (95% CI) or RR[+,-] (95% CI) | Participants (# studies) | Quality of Evidence[c] (GRADE) |
|---|---|---|---|---|
| **Imaging: no evidence** | | | | |

[a]See legend in S3 File for a complete list of non-abbreviated names of all measurement tools.

[b]Data are presented as standardized mean differences (SMD) and 95% confidence intervals (95% CI) unless otherwise indicated (negative SMD favors education materials). Risk ratios are indicated with RR[+] (RR > 1 favors education) and RR[-] (RR < 1 favors education).

[c]Quality of evidence was downgraded for risk of bias,

[1] imprecision,

[2] inconsistency,

[3] indirectness,

[4] publication bias,

[5] or downgraded to very low if there was one study

[6] (more details provided in S3 File).

evidence that PEMs had no effect on fear-avoidance beliefs compared to usual care at immediate-term (2 RCTs, $n$ = 505; SMD = -0.15; 95% CI: -0.33, 0.02; $p$ = 0.09; $I^2$ = 0%), and very low-quality evidence of no effect at short (1 RCT, $n$ = 461; SMD = -0.09; 95% CI: -0.27, 0.09; $p$ = 0.33) and long-term (1 RCT, $n$ = 461; SMD = -0.16; 95% CI: -0.34, 0.02; $p$ = 0.08).

*Stress (n = 1).* We found very low-quality evidence that PEMs were significantly more effective at decreasing stress compared to usual care at long-term (1 RCT, $n$ = 461; SMD = -0.21; 95% CI: -0.39, -0.03; $p$ = 0.02). We found very low-quality evidence that PEMs had no effect on stress compared to usual care at immediate (1 RCT, $n$ = 461; SMD = -0.13; 95% CI: -0.32, 0.05; $p$ = 0.15), short (1 RCT, $n$ = 461; SMD = -0.13; 95% CI: -0.31, 0.06; $p$ = 0.18), and medium-term (1 RCT, $n$ = 461; SMD = -0.15; 95% CI: -0.33, 0.03; $p$ = 0.11).

*Depression (n = 1).* We found very low-quality evidence that PEMs had no effect on depression compared to usual care at immediate (1 RCT, $n$ = 461; SMD = -0.18; 95% CI: -0.36, 0.01; $p$ = 0.06), short (1 RCT, $n$ = 461; SMD = -0.09; 95% CI: -0.27, 0.09; $p$ = 0.35), medium (1 RCT, $n$ = 461; SMD = -0.11; 95% CI: -0.29, 0.07; $p$ = 0.24), and long-term (1 RCT, $n$ = 461; SMD = -0.15; 95% CI: -0.33, 0.03; $p$ = 0.10).

**Patient education materials alone vs. other interventions.** Ten trials [40, 42–47, 52, 56, 59] compared the effect of PEMs to other interventions (Table 2) on LBP-related outcomes for chronic LBP patients. The most commonly measured outcome was pain intensity (n = 10), followed by disability (n = 9), quality of life (n = 5), global improvement (n = 3), anxiety (n = 2), and depression (n = 2). Single studies measured function, pain self-efficacy, fear-avoidance beliefs, catastrophizing, coping, stress, days off work. No studies measured knowledge, attitudes, general beliefs, imaging, physician visits, referrals, or cost. A summary of findings for eight key outcomes are presented in Table 7 (a summary of all other outcomes and forest plots for all analyses are presented in S3 and S4 Files, respectively).

*Pain intensity (n = 10).* We found high-quality evidence that PEMs are less effective than other interventions for decreasing pain intensity at immediate (8 RCT, $n$ = 732; SMD = 0.30; 95% CI: 0.03, 0.56; $p$ = 0.03; $I^2$ = 63%) and short-term (7 RCT, $n$ = 815; SMD = 0.54; 95% CI: 0.20, 0.88; $p$ = 0.002; $I^2$ = 80%). We found moderate and very-low quality evidence that PEMs had no effect on pain intensity compared to other interventions at medium (4 RCT, $n$ = 450; SMD = 0.22; 95% CI: -0.25, 0.69; $p$ = 0.35; $I^2$ = 81%) and long-term (1 RCT, $n$ = 168; SMD = 0.18; 95% CI: -0.12, 0.48; $p$ = 0.24), respectively.

**Table 7. Summary of findings: Education materials compared with another intervention for chronic low back pain.**

| Outcome (# studies) Time points | Outcome measurement tools[a] | SMD[b] (95% CI) or RR[+,-] (95% CI) | Participants (# studies) | Quality of Evidence[c] (GRADE) |
|---|---|---|---|---|
| **Knowledge: no evidence** | | | | |
| **Self-Efficacy (n = 1):** | | | | |
| • Immediate-term (4 wks) | PSEQ (1) | 0.05 [-0.23, 0.33] | 199 (1) | ⊕⊖⊖⊖ Very low[6] |
| • Short-term (12 wks) | PSEQ (1) | 0.06 [-0.22, 0.34] | 199 (1) | ⊕⊖⊖⊖ Very low[6] |
| • Medium-term (24 wks) | PSEQ (1) | 0.04 [-0.24, 0.32] | 199 (1) | ⊕⊖⊖⊖ Very low[6] |
| • Long-term | - | - | 0 (0) | No evidence |
| **Pain (n = 10):** | | | | |
| • Immediate-term (4–8 wks) | SBS (3), VAS (1), NRS (1), BPI (1), PPQ (1), UTs (1) | 0.30 [0.03, 0.56] | 732 (8) | ⊕⊕⊕⊕ High |
| • Short-term (9–12 wks) | NRS (3), SBS (2), BPI (1), UTs (1) | 0.54 [0.20, 0.88] | 815 (7) | ⊕⊕⊕⊕ High |
| • Medium-term (24–26 wks) | SBS (2), BPI (1), UTs (1) | 0.22 [-0.25, 0.69] | 450 (4) | ⊕⊕⊕⊖ Moderate[3] |
| • Long-term (52 wks) | SBS (1) | 0.18 [-0.12, 0.48] | 168 (1) | ⊕⊖⊖⊖ Very low[6] |
| **Disability (n = 9):** | | | | |
| • Immediate-term (4–8 wks) | RMDQ (6), ODI (1) | 0.47 [0.12, 0.83] | 714 (7) | ⊕⊕⊕⊕ High |
| • Short-term (9–12 wks) | RMDQ (6), ODI (2) | 0.64 [0.25, 1.02] | 881 (8) | ⊕⊕⊕⊕ High |
| • Medium-term (24–26 wks) | RMDQ (3), ODI (1) | 0.29 [-0.09, 0.67] | 450 (4) | ⊕⊕⊕⊕ High |
| • Long-term (52 wks) | RMDQ (1) | -0.07 [-0.37, 0.23] | 168 (1) | ⊕⊖⊖⊖ Very low[6] |
| **Quality of Life (n = 5):** | | | | |
| • Immediate-term (4–8 wks) | SF-36 (3), SF-12 (1) | 1.25 [0.14, 2.36]. Two studies did not provide usable data but found no difference between groups | 62 (2) 221 (2) | ⊕⊕⊖⊖ Low[1,2] |
| • Short-term (10–12 wks) | SF-36 (3), SF-12 (1) | 1.01 [-0.99, 3.01]. Two studies did not provide usable data but found (i) no difference between groups or (ii) education to be less effective than other interventions | 228 (2) i. 66 (1) ii. 168 (1) | ⊕⊕⊖⊖ Low[2,3] |
| • Medium-term (26 wks) | SF-36 (1) | One study did not provide usable data but found no difference between groups | 63 (1) | ⊕⊖⊖⊖ Very low[6] |
| • Long-term (52 wks) | SF-12 (1) | One study did not provide usable data but found no difference between groups | 159 (1) | ⊕⊖⊖⊖ Very low[6] |
| **Global Improvement (n = 3):** | | | | |
| • Immediate-term (4–6 wks) | PGIC (1), UTs (1) | 0.53 [0.21, 0.84] | 327 (2) | ⊕⊕⊕⊖ Moderate[2] |
| • Short-term (12 wks) | PGIC (1), UTs (2) | 0.60 [0.16, 1.04] | 509 (3) | ⊕⊕⊕⊕ High |

(*Continued*)

**Table 7.** (Continued)

| Outcome (# studies) Time points | Outcome measurement tools[a] | SMD[b] (95% CI) or RR[+,-] (95% CI) | Participants (# studies) | Quality of Evidence[c] (GRADE) |
|---|---|---|---|---|
| • Medium-term (24–26 wks) | PGIC (1), UTs (1) | 0.55 [0.19, 0.91] | 327 (2) | ⊕⊕⊕⊖ Moderate[2] |
| • Long-term | - | - | 0 (0) | No evidence |
| **Days off work (n = 1):** | | | | |
| • Immediate-term | | - | 0 (0) | No evidence |
| • Short-term (10 wks) | % with days off work (1) | One study did not provide usable data but found no difference between groups | 168 (1) | ⊕⊖⊖⊖ Very low[6] |
| • Medium-term | | - | 0 (0) | No evidence |
| • Long-term | | - | 0 (0) | No evidence |
| **Imaging: no evidence** | | | | |

[a]See legend in S3 File for a complete list of non-abbreviated names of all measurement tools.

[b]Data are presented as standardized mean differences (SMD) and 95% confidence intervals (95% CI) unless otherwise indicated (negative SMD favors education materials). Risk ratios are indicated with RR[+] (RR > 1 favors education) and RR[-] (RR < 1 favors education).

[c]Quality of evidence was downgraded for risk of bias,

[1] imprecision,

[2] inconsistency,

[3] indirectness,

[4] publication bias,

[5] or downgraded to very low if there was one study

[6] (more details provided in S3 File).

*Disability (n = 9).* We found high-quality evidence that PEMs are less effective than other interventions for decreasing disability at immediate (7 RCTs, *n* = 714; SMD = 0.47; 95% CI: 0.12, 0.83; *p* = 0.009; $I^2$ = 79%) and short-term (8 RCT, *n* = 881; SMD = 0.64; 95% CI: 0.25, 1.02; *p* = 0.001; $I^2$ = 85%). We found high and very-low quality evidence that PEMs had no effect on disability compared to other interventions at medium (4 RCT, *n* = 450; SMD = 0.29; 95% CI: -0.09, 0.67; *p* = 0.13; $I^2$ = 72%) and long-term (1 RCT, *n* = 168; SMD = -0.07; 95% CI: -0.37, 0.23; *p* = 0.65), respectively.

*Quality of life (n = 5).* We found low-quality evidence that PEMs were less effective than other interventions for improving quality of life at immediate-term (2 RCTs, *n* = 62; SMD = 1.25; 95% CI: 0.14, 2.36; *p* = 0.03; $I^2$ = 73%). Two studies (2 RCTs; *n* = 221) could not be pooled in the analysis but both found no difference of effect. We found low-quality evidence that PEMs had no effect on quality of life compared to other interventions at short-term (2 RCTs, *n* = 228; SMD = 1.01; 95% CI: -0.99, 3.01; *p* = 0.32; $I^2$ = 96%). Two studies (2 RCTs; *n* = 221) could not be pooled, but one found there to be no difference of effect (*n* = 66), and the other found PEMs to be significantly less effective than other interventions (*n* = 168). Finally, we found very low-quality evidence that PEMs had no effect on quality of life compared to other interventions medium (1 RCT; *n* = 63) and long-term (1 RCT; *n* = 159).

*Global improvement (n = 3).* We found moderate-quality evidence that PEMs are less effective than other interventions on global improvement ratings at immediate (2 RCTs, *n* = 327; SMD = 0.53; 95% CI: 0.21, 0.84; *p* = 0.001; $I^2$ = 22%) and medium-term (2 RCTs, *n* = 327;

SMD = 0.55; 95% CI: 0.19, 0.91; $p$ = 0.003; $I^2$ = 44%), and high-quality evidence of the same observation at short-term (3 RCTs, $n$ = 509; SMD = 0.60; 95% CI: 0.16, 1.04; $p$ = 0.008; $I^2$ = 75%).

*Function (n = 1).* We found very low-quality evidence that PEMs are significantly less effective than other interventions for improving performance-based function measures on the 6-Minute Walk test (1 RCT, $n$ = 19; SMD = 1.34; 95% CI: 0.32, 2.36; $p$ = 0.01) and Sit-to-Stand test (1 RCT, $n$ = 17; SMD = 1.26; 95% CI: 0.18, 2.34; $p$ = 0.02) at immediate-term. We found very low-quality evidence that PEMs had no effect compared to other interventions on the Sit-and-Reach test (1 RCT, $n$ = 19; SMD = 0.95; 95% CI: -0.02, 1.91; $p$ = 0.05) at immediate-term.

*Pain self-efficacy (n = 1).* We found very low-quality evidence that PEMs had no effect on pain self-efficacy compared to other interventions at immediate (1 RCT, $n$ = 199; SMD = 0.05; 95% CI: -0.23, 0.33; $p$ = 0.74), short (1 RCT, $n$ = 199; SMD = 0.06; 95% CI: -0.22, 0.34; $p$ = 0.67), and medium-term (1 RCT, $n$ = 199; SMD = 0.04; 95% CI: -0.24, 0.32; $p$ = 0.77).

*Fear-avoidance (n = 1).* We found very low-quality evidence that PEMs had no effect on fear-avoidance beliefs compared to other interventions at immediate (1 RCT, $n$ = 199; SMD = 0.13; 95% CI: -0.15, 0.41; $p$ = 0.35), short (1 RCT, $n$ = 199; SMD = 0.08; 95% CI: -0.20, 0.36; $p$ = 0.57), and medium-term (1 RCT, $n$ = 199; SMD = 0.00; 95% CI: -0.28, 0.28; $p$ = 1.00).

*Catastrophizing (n = 1).* We found very low-quality evidence that PEMs are significantly less effective than other interventions for reducing catastrophizing thoughts at immediate (1 RCT, $n$ = 199; SMD = 0.50; 95% CI: 0.21, 0.78; $p$ = 0.0006), short (1 RCT, $n$ = 199; SMD = 0.42; 95% CI: 0.14, 0.70; $p$ = 0.003), and medium-term (1 RCT, $n$ = 199; SMD = 0.44; 95% CI: 0.15, 0.72; $p$ = 0.002).

*Coping (n = 1).* We found very low-quality evidence that PEMs had no effect on coping compared to other interventions at immediate (1 RCT, $n$ = 199; SMD = 0.13; 95% CI: -0.14, 0.41; $p$ = 0.34), short (1 RCT, $n$ = 199; SMD = 0.22; 95% CI: -0.05, 0.50; $p$ = 0.12), and medium-term (1 RCT, $n$ = 199; SMD = 0.17; 95% CI: -0.10, 0.45; $p$ = 0.22).

*Anxiety (n = 2).* We found very low-quality evidence that PEMs had no effect on anxiety compared to other interventions at immediate (1 RCT, $n$ = 199; SMD = 0.07; 95% CI: -0.20, 0.35; $p$ = 0.60), and medium-term (1 RCT, $n$ = 199; SMD = 0.13; 95% CI: -0.15, 0.40; $p$ = 0.38), and low-quality evidence of no difference in effect at short-term (1 RCT, $n$ = 199; SMD = 0.65; 95% CI: -0.58, 1.87; $p$ = 0.30; $I^2$ = 88%).

*Stress (n = 1).* We found very low-quality evidence that PEMs had no effect on stress compared to other interventions immediate (1 RCT, $n$ = 199; SMD = 0.17; 95% CI: -0.10, 0.45; $p$ = 0.22) and medium-term (1 RCT, $n$ = 199; SMD = 0.26; 95% CI: -0.02, 0.54; $p$ = 0.07). We found very low-quality evidence that PEMs are significantly less effective than other interventions for decreasing stress at short-term (1 RCT, $n$ = 199; SMD = 0.31; 95% CI: 0.03, 0.59; $p$ = 0.03).

*Depression (n = 2).* We found very low-quality evidence that PEMs had no effect on depression compared to other interventions at immediate (1 RCT, $n$ = 199; SMD = 0.03; 95% CI: -0.25, 0.31; $p$ = 0.84) and medium-term (1 RCT, $n$ = 199; SMD = 0.18; 95% CI: -0.10, 0.46; $p$ = 0.21), and low-quality evidence of no effect at short-term (1 RCT, $n$ = 199; SMD = 0.79; 95% CI: -0.56, 2.14; $p$ = 0.25; $I^2$ = 90%).

*Days off work (n = 1).* We found very low-quality evidence that PEMs had no effect on days off work compared to other interventions at short-term (1 RCT, $n$ = 168). No summary data for this outcome was provided in the study so no point estimate can be provided.

**Intervention vs. intervention + patient education materials (additive effect).** No studies measured the additive effect of PEMs with other interventions.

## Discussion

We found 27 trials that evaluated the effectiveness of PEMs for acute or chronic LBP. Most were at moderate to high risk of bias (most commonly due to insufficient follow-up). We hypothesized that knowledge provided by PEMs would modify beliefs, expectations, and pain self-efficacy, and these changes would positively influence patients' experience or perception of pain, expectations for unnecessary tests or other referrals, and adherence to advice to facilitate recovery compared to those who did not receive PEMs. Compared to usual care for acute LBP, PEMs appear to have at least some positive impacts both for patients and health systems, such as improved short-term pain intensity and immediate-term quality of life. Though the evidence was fairly low quality, knowledge appears to increase with the provision of PEMs across all measured time periods, as well as pain self-efficacy in the short to long-term. For health systems, the evidence was again fairly low quality, but PEMs reduced the short-term number of days off work and long-term physician visits and imaging. Compared to usual care for chronic LBP, PEMs were associated with improved pain intensity, global improvement ratings, and pain self-efficacy across all time periods, and quality of life from short to medium-term with variable levels of very low to moderate evidence. At medium-term, PEMs decreased disability but showed no impact at any other time measurement. The effect of PEMs on fear-avoidance beliefs and stress was more variable: fear-avoidance beliefs decreased in the medium-term, while stress decreased in the long term, with no other measurable impact in the other time periods. PEMs had no impact on depression.

Compared to other interventions, PEMs appear to have limited effectiveness in acute LBP. Though there were only one to two studies in all analyses and the quality of evidence was low to very low, PEMs were less effective in reducing immediate-term pain intensity and the number of long-term days off work and physician visits, with no effect on fear-avoidance beliefs, anxiety, depression, and disability. PEMs showed only a small impact on reducing pain intensity in the medium-term, but not short or long-term. Compared to other interventions in chronic LBP, PEMs had no effect or were less effective for every outcome measured.

### Comparison with existing literature

Though we are the first to assess PEMs alone, our results are supported by and expand on previous literature investigating the effectiveness of patient education for LBP. We found many under-assessed outcomes in the LBP patient education literature, including knowledge. Nevertheless, we did find improvements in knowledge across all measured time points, and we provide the first evidence of effect on this outcome for LBP. Looking to the wider literature, we find similar results for PEMs on knowledge for other conditions like diabetes [66] and cancer [67]. Imaging was another under-assessed outcome measured by only one study in our review. However, we found LBP PEMs can reduce imaging rates, which is consistent with studies where PEMs are used as part of larger multi-component interventions to reduce imaging [68–70].

Our findings differ from those of Traeger et al., [24] who found that individual patient education (with or without PEMs) improved reassurance for acute/subacute LBP. Despite including many of the same studies, we did not find any measures of reassurance. Looking more closely at their methods, we see they combined several proxy outcomes (e.g., anxiety, fear-avoidance, and catastrophizing) as their measure of reassurance. We included these outcomes but analysed them as separate constructs and while many favoured PEMs, they were mostly not statistically significant. This highlights the importance of using validated measures of outcomes.

Our results also expanded on those of Engers et al., [27] who found no studies comparing individual patient education to usual care for chronic LBP. We updated this literature with five recent studies and found PEMs were effective on several clinical and process outcomes. Compared to usual care for acute LBP, they found patient education was significantly more effective in some studies but not others. We had similar findings in this comparison, but since we pooled the results in meta-analyses, we were able to find a trend towards a benefit of PEMs over usual care for most clinical outcomes at most time points. Compared to other interventions, we had similar findings that PEMs had no effect or were less effective for chronic LBP.

## Implications for practice

Our review showed that offering PEMs to patients is preferable to usual care for both acute and chronic LBP. Given that PEMs are relatively inexpensive to produce, easy to provide, and unlikely to cause harm, clinicians may find them an effective adjunct to care. Unfortunately, we could not obtain copies of many of the PEMs that were the focus of the papers in our review despite reaching out to all authors. Additional work will be required to effectively translate these materials into practice and realize their potential.

## Implications for research

Overall, we were disappointed to find that many of the studies included in our review used unvalidated and modified outcome measures (especially for process outcomes) despite the existence of validated measurement tools. This clouds our understanding of the effectiveness of interventions and we recommend that researchers use unmodified, validated tools to measure all outcomes. In addition, many key outcomes were rarely measured (e.g., quality of life, knowledge, pain self-efficacy) or not measured at all (e.g., attitudes, general beliefs). To standardize reporting in clinical trials, we recommend that researchers more frequently assess quality of life as it is a core clinical outcome for LBP alongside pain and disability [71], and suggest developing a similar set of core domains for important process outcomes related to LBP (e.g., fear-avoidance beliefs, catastrophizing, coping, pain self-efficacy) since measures of these outcomes varied substantially across LBP trials. Researchers should work with LBP patients to choose a core set of prioritized, patient-reported outcomes. Finally, PEMs literature lacks adequate reporting on material development as well as measures of intervention adherence and other outcomes related to intervention fidelity, making it difficult to fully understand their effectiveness. We recommend that researchers assess and report these outcomes to determine if the interventions are being provided and received as planned by following intervention reporting guidelines such as the TIDieR checklist [31].

## Future research

PEMs compared to usual care for chronic LBP appear to have more success than those for acute LBP, perhaps because the majority were comprehensive digital interventions (as opposed to the physical booklets most often used for acute LBP) with one or often more of the following: (i) co-development with patients, (ii) text- and video-based information, (iii) instant, tailored feedback based on automated questions, (iv) interactive or gamification components including quizzes and rewards, (v) reminders to use the material and follow recommendations, and (vi) could be accessed anywhere at any time. We recommend future studies compare these newer PEMs to other guideline-recommended interventions (e.g., exercise therapies, massage, CBT) since most studies we found in this comparison used standard physical booklets. Furthermore, most of these studies treated the PEMs group as a control or usual care group,

which may have introduced bias to the comparison and hindered our ability to interpret the results.

### Strengths and limitations

The primary strengths of this review were our adherence to best practices for conducting systematic reviews. We followed all guidance provided in the Cochrane [72] and GRADE [37] handbooks, conducted a sensitive search strategy that adhered to the PRESS guidelines [28], and followed the TIDieR recommendations [31] for reporting of intervention details, which allowed for a more thorough assessment of PEMs. Additionally, we included a comprehensive list of outcomes that are important to all stakeholders, including patients, policymakers, researchers, and clinicians, and compared PEMs to other interventions that are commonly used in practice to provide relative effectiveness. We also sought and obtained additional data from authors who did not report the data within their study. Limitations to this review include the use of unvalidated and modified outcome measures and the conversion of dichotomized data to SMDs where it was necessary to pool the results. Both decisions could have influenced the resulting effect sizes and increased the degree of variability across outcome measures and time periods.

### Conclusion

Due to the degree of variability in the impact of PEMs on all outcomes and across all time periods (likely a result of the heterogeneity of measures and definitions across studies), it is difficult to succinctly and concisely state conclusions for all outcomes. However, it certainly appears that providing PEMs is better than doing nothing (i.e., usual care) as we observed small positive patient and system impacts for both acute and chronic LBP. Given their low cost and relative ease of provision, PEMs appear preferable to usual care, although the quality of evidence is fairly low for this conclusion. Compared to other interventions, PEMs had no effect or were less effective for almost every outcome measured; however, cost effectiveness was not assessed in any of these studies, and it is likely that PEMs were substantially less costly than all other studied interventions. Additionally, in recent years more comprehensive digital PEMs have been developed, and we recommend these are compared to other interventions before making conclusions about their relative usefulness.

### Supporting information

**S1 File. Search strategy.**
(DOC)

**S2 File. Inclusion, exclusion, and GRADE criteria, protocol deviations.**
(DOCX)

**S3 File. Summary of findings (all outcomes and comparisons).**
(DOCX)

**S4 File. Forest plots.**
(DOCX)

**S5 File. PRISMA checklist.**
(DOCX)

## Acknowledgments

We would like to thank Georgia Darmonkow, Anika Shama, and Senem Gözel for their contributions to study screening and data extraction.

## Author Contributions

**Conceptualization:** Bradley Furlong, Holly Etchegary, Kris Aubrey-Bassler, Amanda Hall.

**Formal analysis:** Bradley Furlong.

**Funding acquisition:** Amanda Hall.

**Investigation:** Bradley Furlong, Michelle Swab, Amanda Hall.

**Methodology:** Bradley Furlong, Amanda Hall.

**Validation:** Bradley Furlong, Amanda Hall.

**Writing – original draft:** Bradley Furlong, Andrea Pike, Amanda Hall.

**Writing – review & editing:** Bradley Furlong, Holly Etchegary, Kris Aubrey-Bassler, Michelle Swab, Andrea Pike, Amanda Hall.

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
