## [Decision Letter · Decision Letter 0]

5 Jul 2022

PONE-D-22-16206Patient education materials for non-specific low back pain and sciatica: a systematic review and meta-analysisPLOS ONE

Dear Dr. Furlong,

Thank you for submitting your manuscript to PLOS ONE. After careful consideration, we feel that it has merit but does not fully meet PLOS ONE’s publication criteria as it currently stands. Therefore, we invite you to submit a revised version of the manuscript that addresses the points raised during the review process.

We look forward to receiving your revised manuscript.

Kind regards,

Tariq Jamal Siddiqi

Academic Editor

PLOS ONE

Journal Requirements:

"None declared."

Reviewers' comments:

Reviewer's Responses to Questions

**Comments to the Author**

1. Is the manuscript technically sound, and do the data support the conclusions?

Reviewer #1: Yes

2. Has the statistical analysis been performed appropriately and rigorously? 

Reviewer #1: Yes

3. Have the authors made all data underlying the findings in their manuscript fully available?

Reviewer #1: Yes

4. Is the manuscript presented in an intelligible fashion and written in standard English?

Reviewer #1: Yes

5. Review Comments to the Author

Reviewer #1: The authors presented a meta-analysis of 27 studies evaluating patient education materials for non-specific low back pain and sciatica. The analysis is well-conducted, data is presented in an intelligible manner, and the manuscript has been drafted decently. However, there are a few minor concerns that need to be addressed.

1. In the abstract, the authors have mentioned the results but the relevant numbers are not provided within brackets. Please include the effect sizes within brackets in the abstract as well.

2. In the data extraction section, kindly mention the initials of the reviewers within brackets who have conducted the data abstraction. Also add as to what was done in case there were any discrepancies during screening articles or data extraction. Was a third reviewer consulted or the issues were solved mutually? Please add this too.

3. Review Manager 5 was used for statistical analysis. Can the authors please add the company details of the software within brackets?

4. Kindly cite the reference for Higgins I2 (square) statistic.

5. In the results section, the authors have mentioned the effect sizes and number of RCTs and patients within brackets. However, two important components of meta-analytical results are missing. Kindly add the p values and I2 (square) values within all these brackets as well.

6. PLOS authors have the option to publish the peer review history of their article (what does this mean?). If published, this will include your full peer review and any attached files.

Reviewer #1: No

---

## [Author Response · Author response to Decision Letter 0]

26 Jul 2022

Responses to Reviewer Comments

Reviewer #1: The authors presented a meta-analysis of 27 studies evaluating patient education materials for non-specific low back pain and sciatica. The analysis is well-conducted, data is presented in an intelligible manner, and the manuscript has been drafted decently. However, there are a few minor concerns that need to be addressed.

1. In the abstract, the authors have mentioned the results but the relevant numbers are not provided within brackets. Please include the effect sizes within brackets in the abstract as well.

We agree that adding the effect sizes would add value to the abstract and have now added all the relevant numbers (pg. 2, line 26). Please note that this did require us to exceed PLOS One’s word limit for abstracts.

2. In the data extraction section, kindly mention the initials of the reviewers within brackets who have conducted the data abstraction. Also add as to what was done in case there were any discrepancies during screening articles or data extraction. Was a third reviewer consulted or the issues were solved mutually? Please add this too.

Please note that all data abstraction was performed by BF and doubly abstracted by one of three individuals who have been acknowledged in the acknowledgements section (GD, AS, SG). These contributors only partially contributed to portions of study screening and/or data extraction and do not qualify for authorship. However, we have still added their initials in these sections as requested with reference to the acknowledgements section.

Thank you for your second point. We had already described how screening discrepancies/conflicts would be resolved (i.e., by a third reviewer; please see pg. 5, line 101), but did not do so for the data extraction section. We have now included this (pg. 6, line 108) with the author initials in all sections.

3. Review Manager 5 was used for statistical analysis. Can the authors please add the company details of the software within brackets?

We have added the specific version (5.4.1) and company (The Cochrane Collaboration) details for Review Manager 5 in the manuscript (pg. 7, line 139).

4. Kindly cite the reference for Higgins I2 (square) statistic.

We have cited this paper for the I2 statistic (pg. 8, line 151).

5. In the results section, the authors have mentioned the effect sizes and number of RCTs and patients within brackets. However, two important components of meta-analytical results are missing. Kindly add the p values and I2 (square) values within all these brackets as well.

Thank you for pointing this out, we agree that this will add value to the results section. We have added all relevant p and I2 values to the text (starting on pg. 21, line 227).

---

## [Decision Letter · Decision Letter 1]

24 Aug 2022

PONE-D-22-16206R1Patient education materials for non-specific low back pain and sciatica: a systematic review and meta-analysisPLOS ONE

Dear Dr. Furlong,

Thank you for submitting your manuscript to PLOS ONE. After careful consideration, we feel that it has merit but does not fully meet PLOS ONE’s publication criteria as it currently stands. Therefore, we invite you to submit a revised version of the manuscript that addresses the points raised during the review process.

We look forward to receiving your revised manuscript.

Kind regards,

Tariq Jamal Siddiqi

Academic Editor

PLOS ONE

Journal Requirements:

Reviewers' comments:

Reviewer's Responses to Questions

**Comments to the Author**

1. If the authors have adequately addressed your comments raised in a previous round of review and you feel that this manuscript is now acceptable for publication, you may indicate that here to bypass the “Comments to the Author” section, enter your conflict of interest statement in the “Confidential to Editor” section, and submit your "Accept" recommendation.

Reviewer #1: All comments have been addressed

2. Is the manuscript technically sound, and do the data support the conclusions?

Reviewer #1: Yes

3. Has the statistical analysis been performed appropriately and rigorously? 

Reviewer #1: Yes

4. Have the authors made all data underlying the findings in their manuscript fully available?

Reviewer #1: Yes

5. Is the manuscript presented in an intelligible fashion and written in standard English?

Reviewer #1: Yes

6. Review Comments to the Author

Reviewer #1: The authors have addressed the concerns appropriately. However, there is a minor issue. The authors state, 'Two reviewers (BF, see acknowledgements for AS, SG)'. However, these are a total of 3 reviewers I suppose. Kindly correct this if it is an error.

7. PLOS authors have the option to publish the peer review history of their article (what does this mean?). If published, this will include your full peer review and any attached files.

Reviewer #1: No

---

## [Author Response · Author response to Decision Letter 1]

24 Aug 2022

Responses to Reviewer Comments

Reviewer #1: The authors have addressed the concerns appropriately. However, there is a minor issue. The authors state, 'Two reviewers (BF, see acknowledgements for AS, SG)'. However, these are a total of 3 reviewers I suppose. Kindly correct this if it is an error.

Thank you very much for noticing the wording here. We completely agree that the way we worded this could be confusing for the reader. What we meant to say is that BF performed all actions for all studies, and it was doubly performed by one of the other listed reviewers (e.g., either GD, AS, SG). That is, the work was split up between these three acknowledged individuals so that all screening and data extraction were performed by two reviewers. To clarify, if we had a hypothetical scenario where 1000 studies were found in the search, BF would have screened all 1000 articles, and GD, AS, and SG would have each screened approximately 333 unique articles so that all articles were screened by two reviewers in the end. I hope this makes sense.

We have updated the wording in these instances to make it more explicitly clear:

• pg. 5, line 100 – changed to “(BF, one of GD, AS, SG; see acknowledgements)”

• pg. 6, line 107 – changed to “(BF, one of AS, SG; see acknowledgements)”

---

## [Editor Report · Decision Letter 2]

30 Aug 2022

Patient education materials for non-specific low back pain and sciatica: a systematic review and meta-analysis

PONE-D-22-16206R2

Dear Dr. Furlong,

We’re pleased to inform you that your manuscript has been judged scientifically suitable for publication and will be formally accepted for publication once it meets all outstanding technical requirements.

Kind regards,

Tariq Jamal Siddiqi

Academic Editor

PLOS ONE
---

## [Editor Report · Acceptance letter]

15 Sep 2022

PONE-D-22-16206R2 

Patient education materials for non-specific low back pain and sciatica: a systematic review and meta-analysis 

Dear Dr. Furlong:

I'm pleased to inform you that your manuscript has been deemed suitable for publication in PLOS ONE. Congratulations! Your manuscript is now with our production department. 

Kind regards, 

on behalf of

Dr. Tariq Jamal Siddiqi 

Academic Editor

PLOS ONE